# Temperature Balancing, Layer-wise Weight Analysis, and Neural Network Training

**Yefan Zhou\*** [1], **Tianyu Pang\***[2], **Keqin Liu**[2], **Charles H. Martin**[3], **Michael W. Mahoney**[4]**, and Yaoqing Yang**[1]

[1]Department of Computer Science, Dartmouth College
[2]National Center for Applied Mathematics and Department of Mathematics, Nanjing University
[3]Calculation Consulting
[4]ICSI, LBNL, and Department of Statistics, University of California at Berkeley
{yefan.zhou.gr, yaoqing.yang}@dartmouth.edu
tianyupang628@gmail.com, kqliu@nju.edu.cn
charles@CalculationConsulting.com, mmahoney@stat.berkeley.edu

## Abstract

Regularization in modern machine learning is crucial, and it can take various forms in algorithmic design: training set, model family, error function, regularization terms, and optimizations. In particular, the learning rate, which can be interpreted as a temperature-like parameter within the statistical mechanics of learning, plays a crucial role in neural network training. Indeed, many widely adopted training strategies basically just define the decay of the learning rate over time. This process can be interpreted as decreasing a temperature, using either a global learning rate (for the entire model) or a learning rate that varies for each parameter. This paper proposes `TempBalance`, a straightforward yet effective layer-wise learning rate method. `TempBalance` is based on Heavy-Tailed Self-Regularization (HT-SR) Theory, an approach which characterizes the implicit self-regularization of different layers in trained models. We demonstrate the efficacy of using HT-SR-motivated metrics to guide the scheduling and balancing of temperature across all network layers during model training, resulting in improved performance during testing. We implement `TempBalance` on CIFAR10, CIFAR100, SVHN, and TinyImageNet datasets using ResNets, VGGs and WideResNets with various depths and widths. Our results show that `TempBalance` significantly outperforms ordinary SGD and carefully-tuned spectral norm regularization. We also show that `TempBalance` outperforms a number of state-of-the-art optimizers and learning rate schedulers.

## 1 Introduction

Having a learning rate schedule that gradually decreases over time is crucial for the convergence and performance of state-of-the-art machine learning algorithms. Indeed, many optimization algorithms essentially boil down to designing a progression of parameter updates, as realized by different learning rate schedules [1–4]. Common schedules assign a global learning rate per epoch, where the same learning rate is used for all layers in the model. This includes the family of cyclical learning rates [3], and parameter-wise learning rate schedules like Adam [2] and its variants [5, 6]. However, such a global learning rate schedule does not take into account the structural characteristics of neural networks (NNs). At the same time, parameter-wise learning rate schedules are sometimes used, but they have long been conjectured to have worse generalization performance than carefully tuned

---

*First two authors contributed equally.

37th Conference on Neural Information Processing Systems (NeurIPS 2023).

stochastic gradient descent (SGD) optimizers [7], and storing both first and second-order moments for each parameter can lead to substantially increased memory consumption [8]. As mentioned in [9], storing the whole Megatron-Turing NLG requires 10 terabytes of aggregate memory, and the Adam optimizer's first and second-order moments [2] consume 40% of the memory. Nonetheless, improving parameter-wise learning rate schedules is an active field of study [4, 5, 10, 11].

A largely under-explored idea to reconcile the two extremes of setting a single global learning rate or assigning fine-grained parameter-level learning rates is to assign layer-wise learning rates. Such a learning rate assignment method does not require much storage cost, and it can assign very different training speeds to different layers. However, existing layer-wise schemes are often introduced as an additional part of hyperparameter sweeping, thus substantially increasing computational cost; and most lack a strong (or any) theoretical foundation. For instance, layer-wise learning rates can increase test accuracy in transfer learning [12] and domain adaptation [13], but these learning rates are often empirically tuned. More recently, motivated by the intuition that lower-level layers should be domain-specific and higher-level layers should be task-specific, [14] automates the search for an optimal set of learning rates. However, the authors find the nested, bi-level optimization scheme to be too computationally expensive in practice [15]. AutoLR also automatically tunes its layer-wise learning rates according to the "role" of each layer [16]. The method is validated almost entirely by empirical results, further explained by layer-wise weight variations. While the authors attempt to assign a different initial learning rate to each layer, the learning rate for each layer continues to stay largely constant throughout training. LARS [17, 18] is another method to assign layer-wise learning rate. It is based on the "trust ratio," defined as the ratio of weight norm to gradient update norm of each layer, and it is specifically used in large batch training to avoid gradient divergence.

In this paper, we propose TempBalance, a simple yet effective layer-wise learning rate assignment (and regularization) method, grounded in Heavy-Tail Self Regularization (HT-SR) Theory [19–24]. Our approach leverages HT-SR Theory to assess the quality of each network layer. This is achieved through an analysis of the heavy-tail (HT) structure present in the Empirical Spectral Density (ESD) of NN weight matrices. Given this information, TempBalance meticulously adjusts the *temperature-like parameter* to control each layer's quality, with the objective of ensuring consistently high quality across all layers of the network. From the *statistical physics viewpoint* on learning and optimization [24–28], a temperature-like parameter refers to some quantity related to the empirical noise/stochasticity of the learning process. This is the noise scale described by [29, 30], and it can be written as a function of learning rate, batch size, and momentum. Prior research [19, 31] has shown that temperature-like parameters significantly influence HT structure in the ESD. Our approach, TempBalance, focuses on the strategic adjustment of the learning rate as the temperature-like parameter, thereby facilitating accurate control of the quality across each network layer, as characterized by its HT ESD structure. The following paragraph will delve deeper into the importance of HT-SR, highlighting its connection to the concept of layer-wise temperature.

**HT-SR Theory.** HT-SR Theory [19–24] relies on the empirical fact that very well-trained models tend to exhibit strong correlations, resulting in HT structure in the ESD of each layer. To obtain this ESD, we take a NN with $L$ layers and the corresponding weight matrices $\mathbf{W}_1, \mathbf{W}_2, \cdots, \mathbf{W}_L$ with shape $n \times m$ (where $n \leq m$). For the $i$-th layer, we calculate the eigenvalues of its correlation matrix $\mathbf{X}_i = \mathbf{W}_i^T \mathbf{W}_i$, and then we plot the ESD for that layer. Upon training, the ESD will typically gradually change to have an HT structure [19, 22]. We can then fit a power law (PL) distribution to the HT part of the ESD, and extract its exponent as, namely, PL_Alpha. The fitted PL will have the following formula:

$$p(\lambda) \propto \lambda^{-\alpha}, \quad \lambda_{\min} < \lambda < \lambda_{\max}. \tag{1}$$

The PL_Alpha metric measures the PL exponent of the weight matrices' ESD. Its underlying motivation stems from random matrix theory and statistical physics, as well as the empirical observation that HT ESDs are ubiquitous in well trained NN models [20, 22].

The PL_Alpha metric has been shown to predict the trends in the test accuracy of state-of-the-art models in computer vision (CV) and natural language processing (NLP), without even the need for access to training or testing data [22, 32]. According to [22], one can aggregate PL_Alpha's for different layers either by simple averaging or weighted averaging, and each can predict test accuracy in different cases [22, 32]. Furthermore, the *layer-wise* nature of PL_Alpha makes it a fine-grained metric that can be used to assess the quality of individual layers of the network. Thus, in this paper, we extend and apply HT-SR Theory (originally designed as a predictive diagnostic for analyzing

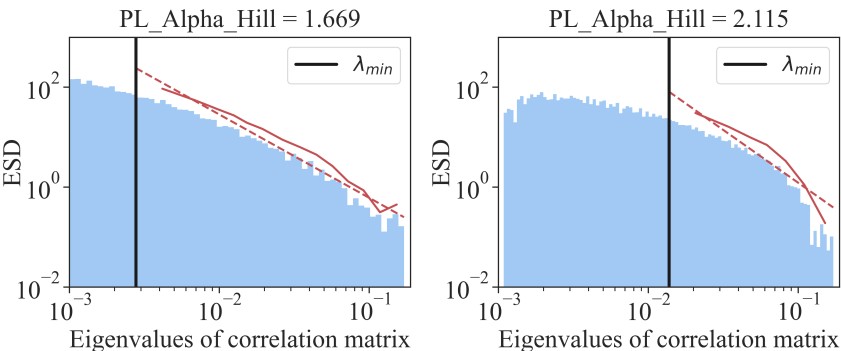

Figure 1: Examples of PL fitting. Blue histograms depict the ESDs. Vertical black lines indicate the lower threshold $\lambda_{\min}$ used to truncate the full ESDs and extract the tail portion. Solid red curves represent the tail part of the ESDs truncated by $\lambda_{\min}$, while dashed red curves represent the fitted HT distributions. The left shows a more HT ESD, which requires a relatively lower learning rate. The right one shows a less HT ESD, which requires a relatively higher learning rate. Unlike prior work, we do not aim to find the "optimal" PL exponent. (Thus, we are less interested in obtaining a precise estimate than in obtaining a robust estimate.) Instead, we use the PL exponent to *rank* ESDs to find layers that need higher/lower learning rates. These two ESDs correspond to two layers of a ResNet18 model trained on TinyImageNet.

pre-trained NN models) to NN training, and we exploit the layer-wise information provided by `PL_Alpha` to determine the layer-wise learning rates for better test accuracy.

We note that, while it provides perhaps the most principled approach, the `PL_Alpha` metric is not the only way to try to measure the HT structure in NN models. Several recent papers [33–35] use different HT metrics to try to measure the spectral properties of other matrices (such as input/output covariance matrices, Fisher Information Matrices, and the Hessian). We show in Appendix A that these HT phenomena, measured in different ways on different matrices, are closely related to each other. On the other hand, this also means that (for the problems considered in this paper) the absolute numerical value of `PL_Alpha` is less important, as optimal PL exponents estimated by different algorithms can be different [22, 33]. What matters the most, as we show in this paper, is the layer-wise quality *ranked* by the PL exponent: layers with a smaller `PL_Alpha` tend to be relatively more "overtrained," and layers with a larger `PL_Alpha` tend to be relatively more "undertrained." (We emphasize that this is true for the training problem we consider in this paper—for prior HT-SR work, the actual numerical value of `PL_Alpha` mattered *a lot*.)

This observation leads to a simple and efficient way to *balance* layer-wise learning rates: assign a lower learning rate to more overtrained layers and a larger learning rate to more undertrained layers, using `PL_Alpha` (see Figure 1). In implementing this learning rate balancing approach, we use a *scale-free* method to map the `PL_Alpha` value of each layer to a predefined learning rate range. This range is established in relation to a *global* learning rate. Rather than depending on the absolute numerical values of `PL_Alpha` for each layer, this method emphasizes the importance of their relative differences and quality ranking. As a result, the learning rates assigned to individual layers remain stable and unaffected by arbitrary linear scaling of `PL_Alpha` estimates, whether they arise from the choice of the estimator or the presence of noisy measurements. On top of this, we can perform a grid search on the global learning rate. This is standard practice, and it is more efficient than grid-searching the layer-wise learning rates. We use this combination of assigning layer-wise learning rates using `PL_Alpha` and grid-searching the base global learning rate to avoid having to decide the "optimal" PL exponent, as this can be tricky due to different ways of measuring HT properties. Indeed, there are different ways to measure `PL_Alpha` [19], and we use the Hill estimator [36]. While not necessarily the best estimate (see [19, 22]), it shows stable performance in our experiments. We refer to our version of the `PL_Alpha` metric as the `PL_Alpha_Hill` metric, and we use it for the remainder of the paper.

Another popular way to change the ESD of weights is to constrain the spectral norm (i.e., the largest eigenvalue) using spectral norm regularization (`SNR`) [37, 38]. `SNR` provides a different form of regularization, compared to HT-SR, because it regulates the largest eigenvalue instead of the ESD slope (i.e., the `PL_Alpha_Hill` metric). It has been demonstrated that the spectral norm

and `PL_Alpha_Hill` serve distinct roles in evaluating model quality, and their combined form yields optimal predictions for test accuracy trends [19, 22, 23, 32]. To complement this, our results demonstrate that `TempBalance` outperforms `SNR` in training deep NNs in most cases. Moreover, when these two regularization methods are combined during training, they result in optimal test accuracy, thereby confirming their complementary roles. As described in [23, 32], the spectral norm and `PL_Alpha_Hill` measure the scale and the shape of a ESD, respectively; and regulating both the scale and shape is crucial for achieving better ESD regularization. We provide ablation studies on several layer-wise metrics for assigning layer-wise learning rates, including spectral norm, and we show that `PL_Alpha_Hill` performs the best among them.

**Our main contributions.** The following summarizes our main contributions.[2]

- We propose a simple yet effective layer-wise learning rate schedule, `TempBalance`, which is motivated by HT-SR Theory. Based on our empirical results, we obtain two main high-level insights. First, the mapping from `PL_Alpha_Hill` to learning rates should be scale-free, meaning that arbitrary linear scaling on the estimated PL exponent should not change the learning rate assignment. Second, searching for the minimum eigenvalue $\lambda_{\min}$, a standard practice in PL fitting [19, 39, 40], leads to unstable training. To improve stability, we instead fix $\lambda_{\min}$ as the median of the ESD.
- We compare `TempBalance` to ordinary `SGD` and `SNR` on various training tasks. This includes (1) different network architectures, such as ResNet, VGG, WideResNet, (2) different datasets, such as CIFAR10, CIFAR100, SVHN, TinyImageNet, and (3) ablation studies, such as varying widths, depths, initial learning rates, HT-SR layer-wise metrics, and PL fitting methods. Compared to ordinary `SGD`, `TempBalance` achieves higher test accuracy by setting layer-wise learning rates. Compared to `SNR`, `TempBalance` performs better by providing a more fine-grained regularization on *shape/slope* instead of norm. We also show that combining `TempBalance` and `SNR` leads to further improved accuracy, verifying their complementary roles in informing deep learning training.
- We compare `TempBalance` to a range of state-of-the-art optimizers and learning rate schedulers, including SGDR [10], SGDP [41], `Lookahead` [42] and `LARS` [17, 18] on ResNet18 and ResNet34 trained on CIFAR100. We show that `TempBalance` achieves the highest test accuracy. We do careful hyperparameter tuning for all baselines. All results are obtained from five random seeds.
- We use ablation studies to show that `PL_Alpha_Hill` provides the best test accuracy among several layer-wise metrics considered by HT-SR [22, 32]. We also show that `TempBalance` maintains stable performance over `SGD` baselines when the model size changes. Furthermore, we show visualization results in Appendix B, verifying that `TempBalance` controls ESDs during training.

## 2 Related Work

Here, we give an overview of the statistical mechanics of learning and recent progress in theoretical and empirical studies on generalization metrics and their applications.

**Statistical mechanics of learning and HT-SR.** Our paper is motivated by statistical mechanics of learning [43–45], and especially by works that connect load-like [43, 46, 47] and temperature-like parameters [25, 48] to NNs. According to prior works in this area [24, 49], a *temperature-like parameter* represents the amount of noise/variance in an iteration of `SGD`, such as learning rate, weight decay parameters, and batch size. A *load-like parameter* represents the quantity and/or quality of data relative to the size of the learning model. To measure the quality of publicly-available pre-trained NNs, Martin and Mahoney [19] introduce HT-SR Theory, showing that the weight matrices of deep NNs exhibit HT ESDs, and they show that a decay coefficient of ESD, `PL_Alpha`, effectively gauges model quality. Subsequently, [31, 50–54] provide rigorous bounds for HT phenomenon and generalization, further adding support to HT-SR Theory. HT-SR has also been applied to predicting trends in test accuracy of large-scale NNs, in both CV and NLP [22, 23, 32], but it has yet to be systematically incorporated to novel training algorithms. Recently, more papers realize the important connections between deep NNs and statistical mechanics of learning [24]. To name a few, Yang et al. [49] use load and temperature parameters to study a wide range of loss landscapes, providing a taxonomy from the perspective of global structure of a loss landscape. On the theory side, Baity-Jesi et al. [55] investigates the glassy behavior of NNs, and Barbier et al. [56] derives the optimal generalization error of generalized linear systems. More recently, Sorscher et al. [57] studies easy versus hard samples used in training and design a "data-pruning" method; and Zhou et al. [58] establishes a

---

[2]Our code is open-sourced: https://github.com/YefanZhou/TempBalance.

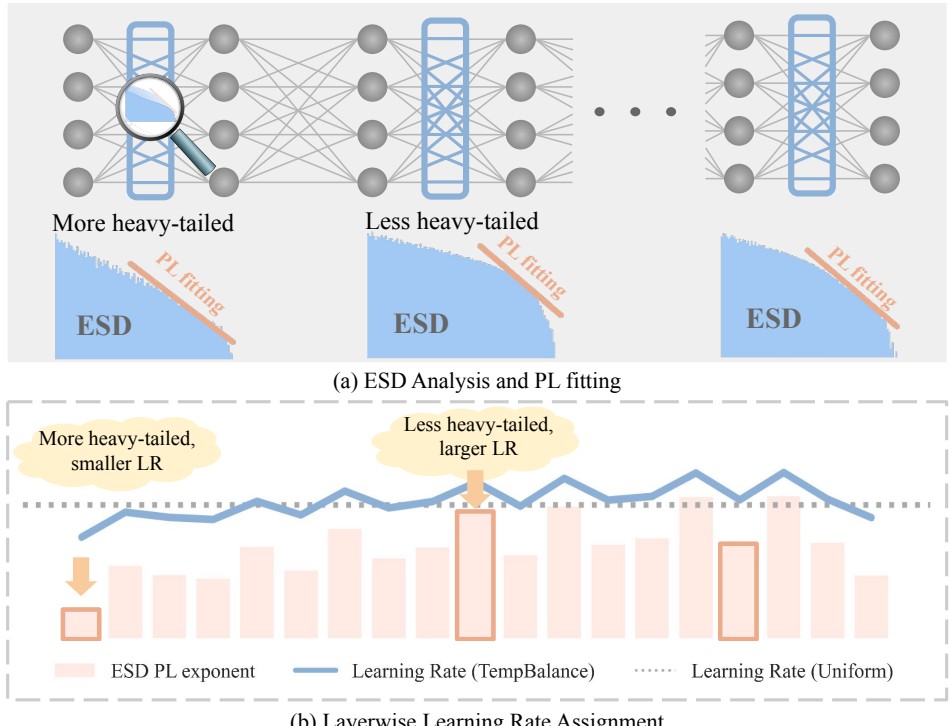

(a) ESD Analysis and PL fitting

(b) Layerwise Learning Rate Assignment

Figure 2: The pipeline diagram of `TempBalance`. In each epoch, `TempBalance` undergoes two steps: (a) Performing ESD analysis on all layers and employing PL fitting to derive the layer-wise `PL_Alpha_Hill`, and (b) Using the layer-wise `PL_Alpha_Hill` to assign learning rates to each layer using an assignment function.

"three-regime model" in network pruning, unifying multiple practical hyperparameter tuning methods in a principled way.

**Generalization measures.** The search for effective and robust generalization metrics (which, importantly, can be very different than model quality metrics [32]) has been the focus of several recent theoretical and empirical works [22, 32, 49, 59–61]. Several recent papers apply metric-informed training and architecture search, such as those based on the Hessian [4, 62–64], spectral norm [37, 38], stable rank [65], and the spectrum of the neural tangent kernel [66]. However, most generalization metrics, such as those based on the PAC-Bayes bounds [67–70], do not straightforwardly transfer to layer-wise quality metrics, because such generalization metrics often study the whole NN as an architecture-free function, and they lack the fine granularity to unveil the quality of each layer. Also, it has been mentioned in the literature [59] that (1) directly regularizing generalization metrics can lead to difficulty in training, (2) evaluating these regularization methods may be hard due to the existence of *implicit regularization* in SGD, and (3) these metrics, especially norm-based metrics, cannot be expected to correlate with test accuracy causally [60], making the link between these generalization metrics and practical training methods nuanced. It will be clear in the next section that we do not regularize ESD metrics directly. Instead, we change learning rates to modify ESDs.

## 3 The `TempBalance` Algorithm

In this section, we introduce our simple yet effective method `TempBalance`, based on the `PL_Alpha_Hill` metric from HT-SR Theory. For a NN, different layers tend to have different values for `PL_Alpha_Hill`, [19, 24]: a layer with a larger `PL_Alpha_Hill` indicates that layer is relatively undertrained, while a layer with a smaller `PL_Alpha_Hill` indicates that layer is relatively overtrained. A natural idea is to adjust the degree of learning among different layers to get a balance: for a layer whose `PL_Alpha_Hill` is too large, we could assign a larger learning rate to accelerate its learning, and vice versa. The intuition of our method is transferring one layer's learning rate to another and hence, `TempBalance`. The pipeline is in Figure 2.

**Algorithm 1:** `TempBalance`

---

**Input:** $M$: Deep NN,    $T$: Total training epoch,    $t$: Current epoch,    $\alpha_t^i$: $i_{\text{th}}$ layer's `PL_Alpha_Hill` at epoch $t$,    $\eta_t$: Baseline global learning rate at epoch $t$, $s_1, s_2$: Minimum and maximum scaling ratio,    $f_t$: Learning rate schedule function

**1**   Initialize model $M$;
**2**   **for** $t \leftarrow 0$ *to* $T$ **do**
**3**      Compute $\alpha_t^i$ for all layers using the Hill estimator;
**4**      Leverage all $\alpha_t^i$ and adopt $f_t$ in (2) to assign per-layer learning rate $f_t(i)$ between $s_1\eta_t$ and $s_2\eta_t$ for the next epoch;
**5**      Update the optimizer for the next epoch;
**6**   **end**

---

We provide the details of `TempBalance` in Algorithm 1. Based on `PL_Alpha_Hill` in different layers, we use the learning rate schedule function $f_t$ to map the $i$-th layer to a particular learning rate $f_t(i)$ in epoch $t$. We adopt $f_t$ as a linear map between the layer-wise `PL_Alpha_Hill` and the final layer-wise learning rate, which has the following formula:

$$f_t(i) = \eta_t \cdot \left[ \frac{\alpha_t^i - \alpha_t^{\min}}{\alpha_t^{\max} - \alpha_t^{\min}} (s_2 - s_1) + s_1 \right], \tag{2}$$

where $\eta_t$ means the base global learning rate in epoch $t$, $(s_1, s_2)$ are the minimum and maximum learning rate scaling ratio relative to $\eta_t$, $\alpha_t^i$ represents the layer $i$'s `PL_Alpha_Hill` at the beginning of epoch $t$, and $(\alpha_t^{\min}, \alpha_t^{\max})$ denote the minimum and maximum `PL_Alpha_Hill` across all the layers in epoch $t$. Using (2), we ensure that the new learning rate $f_t(i)$ is a scaled version of the original base learning rate $\eta_t$ and is always inside the interval $[s_1\eta_t, s_2\eta_t]$. Note that $(s_1, s_2)$ serves as tunable hyperparameters in our method. We conducted ablation studies on it, which are detailed in Appendix C. The hyperparameter values used across all experiments can be found in Appendix D. Our studies reveal that the optimal results are usually achieved around $(0.5, 1.5)$.

To fit the PL distribution $p(\lambda)$ defined in (1), we use the Hill estimator [36, 71]. (It is not the best estimator for fine-scale diagnostics based on HT-SR Theory [19, 22], but it is robust, and it suffices for our purposes.) For the $i$-th layer, suppose the weight matrix is $\mathbf{W}_i$ and the correlation matrix $\mathbf{W}_i^\top \mathbf{W}_i$ has ascending eigenvalues $\{\lambda_i\}_{i=1}^n$. Then, the Hill estimator calculates `PL_Alpha_Hill` using the following:

$$\texttt{PL\_Alpha\_Hill} = 1 + \frac{k}{\left( \sum_{i=1}^k ln \frac{\lambda_{n-i+1}}{\lambda_{n-k}} \right)}, \tag{3}$$

where $k$ is the adjustable parameter. We adopt $k = \frac{n}{2}$ in our experiments. Note that changing $k$ essentially changes the lower eigenvalue threshold $\lambda_{\min}$ for (truncated) PL estimation, as shown by the vertical black line in Figure 1. Choosing $k = \frac{n}{2}$ means using the largest half of the eigenvalues to estimate the slope. We empirically find that fixing $k$ for all layers leads to more stable performance than searching $k$ for different layers (e.g., optimizing $k$ using the Kolmogorov–Smirnov test [40], as is needed for other applications of HT-SR Theory [19, 22]).

One advantage of mapping `PL_Alpha_Hill` to learning rates using (2) is that the scale of `PL_Alpha_Hill` is unimportant, i.e., linearly scaling `PL_Alpha_Hill` arbitrarily does not change the learning rate assignment because the linear scaling cancels each other in (2). This can maximally reduce the artifact of estimating the ESD PL exponent/slope due to estimation noise, which has been found to be a tricky issue in practice [19, 23].

## 4   Empirical results

In this section, we give full details of the experimental setup (Section 4.1) and compare our method `TempBalance` to a few baselines (Section 4.2), and then (Section 4.3) we perform ablation studies on varied initial learning rates, model widths, HT-SR layer-wise metrics, and PL fitting methods.

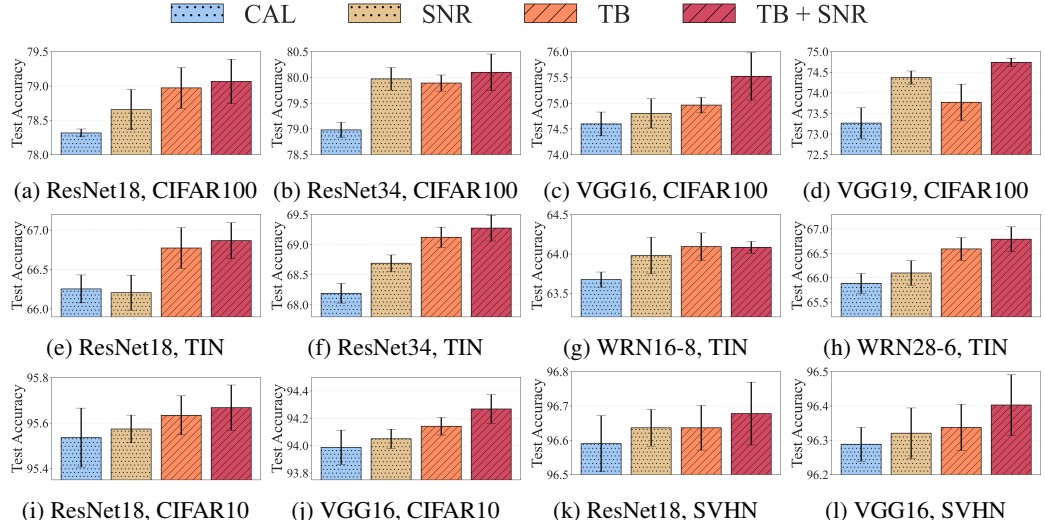

Figure 3: **(Main result).** Comparing our method `TempBalance` (TB) to `CAL` and `SNR`. Our method `TempBalance` outperforms `CAL` and `SNR` in almost all the settings except for VGG19 and ResNet 34 on CIFAR 100. For all experiments, combining `TempBalance` and `SNR` (TB+SNR) yields the best performance. All baselines are carefully tuned. All results are obtained by running five random seeds. See Appendix D for the details in all hyperparameters.

## 4.1 Experimental setup

**Datasets.** We consider CIFAR100, CIFAR10, SVHN and Tiny ImageNet (TIN) [72–75]. CIFAR100 consists of 50K pictures for training and 10K pictures for testing with 100 categories. CIFAR10 consists of 50K pictures for training and 10K pictures for testing with 10 categories. SVHN consists of around 73K pictures for training and around 26K pictures for testing with 10 categories. Tiny ImageNet consists of 10K pictures for training and 10K images for testing with 200 classes.

**Models.** We mainly consider three types of NNs: VGG, ResNet, and WideResNet (WRN) [76–78]. For each network, we consider two different size options. For VGG, we consider VGG16 and VGG19. For ResNet, we consider ResNet18 and ResNet34. For WideResNet, we consider WRN16-8 and WRN28-6. Also, for ResNet and VGG, we consider three different widths for ablation studies.

**Hyperparameters.** One baseline is ordinary `SGD` training with a cosine annealing learning rate schedule (`CAL`), which follows the formula: $\eta_t = \frac{\eta_0}{2}\left(1 + \cos\left(\frac{t\cdot\pi}{T}\right)\right)$, where $t$ is the current epoch, $T$ represents the total training epochs, and $\eta_0$ is the initial learning rate. We grid search the optimal initial (base) learning rate $\eta_0$ for the `CAL` baseline, using the grid $\{0.05, 0.1, 0.15\}$ for ResNet and $\{0.025, 0.05, 0.1\}$ for VGG. The momentum and weight decay are 0.9 and $5 \times 10^{-4}$, respectively, which are both standard choices.

Another baseline is spectral norm regularization (`SNR`). Prior work [37] uses the `SNR` objective:

$$\min_{\Theta}\frac{1}{n}\sum_{i=1}^{n} l\left(f_{\Theta}\left(\boldsymbol{x}_i\right), \boldsymbol{y}_i\right) + \frac{\lambda_{sr}}{2}\sum_{l=1}^{L}\sigma\left(W_l\right)^2, \qquad (4)$$

where $\lambda_{sr}$ is the `SNR` coefficient, $\sigma(W_l)$ is the largest eigenvalue, i.e., spectral norm of weight matrix $\mathbf{W}_l$, and $L$ is the number of layers. We use the power iteration method to calculate $\sigma(W_l)$ in our experiments. For `SNR`, we grid search the optimal regularization coefficient $\lambda_{sr}$, and we again adopt the `CAL` schedule for `SNR`, similar to the `CAL` baseline.

To make our results fully reproducible, we report in Appendix D all hyperparameters, random seeds, and all numerical values of experimental results shown in the figures.

## 4.2 Comparing `TempBalance` and multiple baseline methods.

First, we compare `TempBalance` to two baseline training methods. See results in Figure 3. In the figure, `CAL` means `SGD` training with a `CAL` learning rate schedule, and `SNR` means `SGD` trained

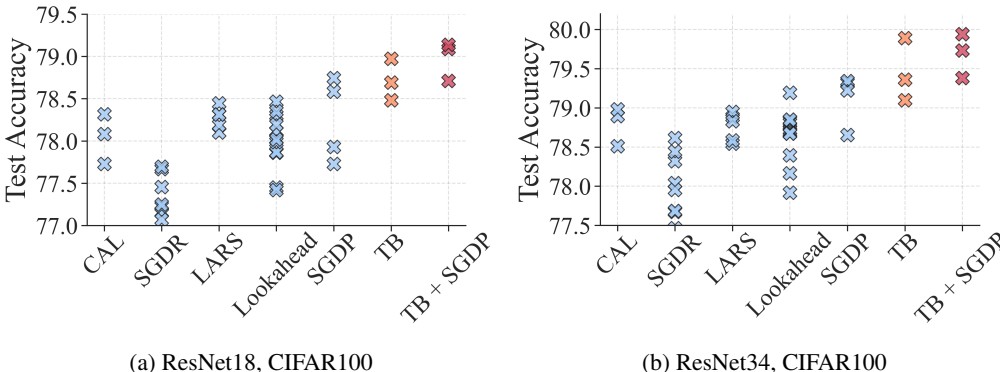

(a) ResNet18, CIFAR100                    (b) ResNet34, CIFAR100

Figure 4: **(More baseline optimizers).** Comparing our method `TempBalance` (TB) to cosine anneal-ing (`CAL`) baseline and other state-of-the-art optimizers and learning rate schedulers for ResNet18 and ResNet34 trained on CIFAR100. Crosses for the same method represent different hyperparameter settings. Each cross represents the mean test accuracy of five random seeds. The best performing model thus far is TB combined with SGDP.

with spectral norm regularization. TB means our method `TempBalance`, and TB + SNR means `TempBalance` combined with SNR. All error bars are obtained from five random seeds. From Figure 3, we see that `TempBalance` outperforms the `CAL` baseline in all settings. In almost all cases, it performs better than SNR baseline. When `TempBalance` does not outperform SNR, combining SNR with `TempBalance` leads to better test accuracy.

Second, we compare our method to a number of optimizers and learning rate schedulers that are not necessarily related to ESD of weights. These include SGDR [10], SGDP [41], `Lookahead` [42] and `LARS` [17, 18], and we compare these baselines with `TempBalance` for ResNet18 and ResNet34 trained on CIFAR100. SGDR is stochastic gradient descent with warm restarts. SGDP modifies the ordinary SGD to compensate for the effect of increasing weight norm. `Lookahead` [42] modifies SGD by letting each gradient update approximate the future trajectory of multiple updates. `LARS` assigns layer-wise learning rates based on the so-called "trust-ratio" and is the closest to our method. Results in Figure 4 show that `TempBalance` outperforms these baselines, and `TempBalance` combined with SGDP is the best-performing method. The crosses on each column represent training runs with different hyperparameters. Note that there are several other methods based on modifying the `Adam` optimizer [2], such as `AdamW` [11], `AdamP` [41] and `LAMB` [79]. However, we do not find them to provide better results than the SGD baseline with cosine annealing (`CAL` in Figure 4). The results are detailed in Appendix E.

### 4.3 Corroborating results and ablation studies.

In addition to the main results (Figures 3 and 4), we provide corroborating results and ablation studies.

**Experiment one: tuning initial learning rate** $\eta_0$. We train models from scratch using `TempBalance` versus `CAL` with various initial learning rates, comparing `TempBalance` and the `CAL` baseline when both methods are allowed to search for the optimal hyperparameters. We again use ResNet18, ResNet34, VGG16 and VGG19 as our architectures and show results on CIFAR100. Results in Figure 5 show that `TempBalance` achieves a higher test accuracy than `CAL` for both ResNet and VGG.

**Experiment two: varying channel width.** We view the fraction of model width in Experiment one as "100%," and we experiment with models with varied widths in $[50\%, 100\%, 150\%]$. We again used ResNet18, ResNet34, VGG16 and VGG19, and trained on CIFAR100, and we grid search for the optimal learning rate for each width to get the best accuracy. Results in Figure 6 show we find that `TempBalance` outperforms the baseline for all widths.

**Experiment three: varying HT-SR metric**. We use different HT-SR metrics to assign layer-wise learning rates. That is, we replace the layer-wise `PL_Alpha_Hill` in (2) with other HT-SR metrics including `SpectralNorm` and `AlphaWeighted` [22]. Results in Figure 7 show that `PL_Alpha_Hill` achieves the optimal test accuracy.

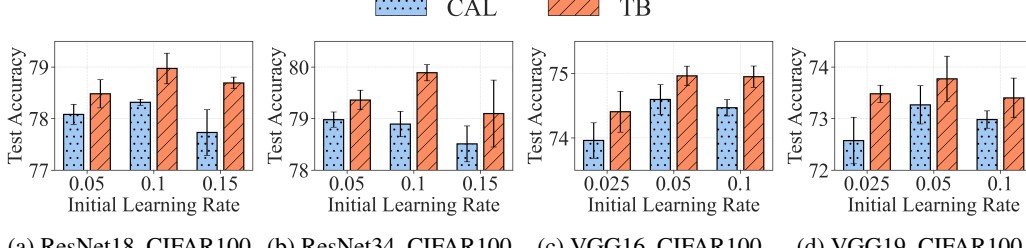

(a) ResNet18, CIFAR100  (b) ResNet34, CIFAR100  (c) VGG16, CIFAR100  (d) VGG19, CIFAR100

Figure 5: **(Tuning initial learning rate).** Comparing the test accuracy of `TempBalance` (red) and `CAL` baseline (blue) for varying initial learning rate. Our method `TempBalance` outperforms `CAL` for both ResNet and VGG trained on CIFAR100. All results are obtained by running five random seeds.

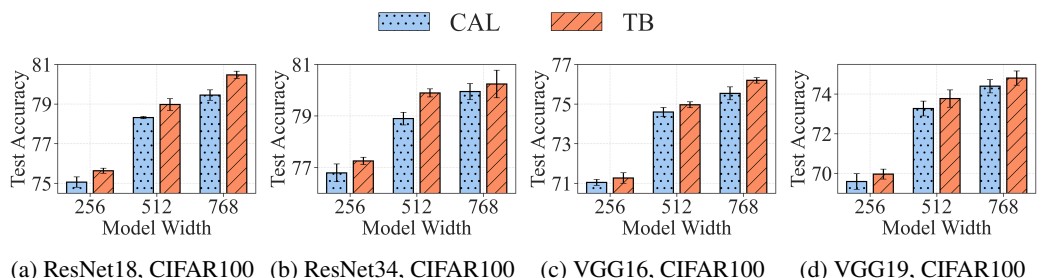

(a) ResNet18, CIFAR100  (b) ResNet34, CIFAR100  (c) VGG16, CIFAR100  (d) VGG19, CIFAR100

Figure 6: **(Different widths).** Comparing `TempBalance` and the `CAL` baseline for different network widths. Our method `TempBalance` consistently outperforms the `CAL` baseline across various network widths for both ResNet and VGG trained on CIFAR100. All results are obtained by running five random seeds.

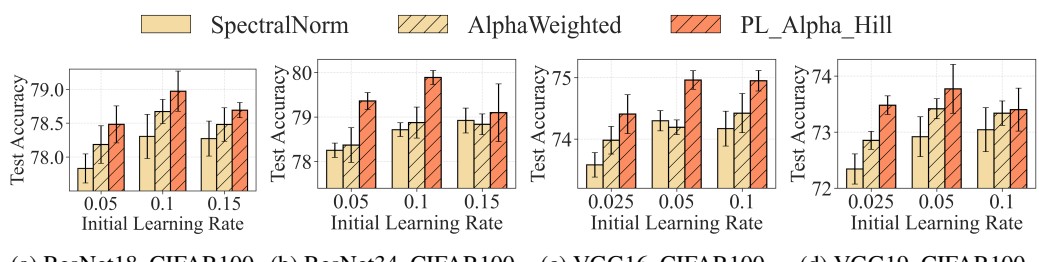

(a) ResNet18, CIFAR100  (b) ResNet34, CIFAR100  (c) VGG16, CIFAR100  (d) VGG19, CIFAR100

Figure 7: **(Different HT-SR metrics).** Comparing `PL_Alpha_Hill` with multiple HT-SR metrics. `PL_Alpha_Hill` achieves the best test accuracy among these metrics. All results are obtained by running five random seeds.

**Experiment four: varying PL fitting methods.** The HT-SR metric `PL_Alpha_Hill` is derived through PL fitting, which is influenced by the choice of hyperparameter $\lambda_{\min}$. More specifically, this involves determining the adjustable parameter $k$ as per Equation 3. Past research has employed various methods to select $\lambda_{\min}$ based on the task, such as performance prediction. For instance, Martin et al. [22], Clauset et al. [39] choose $\lambda_{\min}$ that aligns with the best fit according to the Kolmogorov-Smirnov statistic [40], a method termed `Goodness-of-fit`. Meanwhile, Yang et al. [32] adopted the `Fix-finger` approach, which identifies $\lambda_{\min}$ at the peak of the ESD. In our study, we designate $\lambda_{\min}$ as the median of all eigenvalues present in the ESD for `TempBalance`. As depicted in Figure 8, our fitting method, termed `Median`, not only ensures optimal test accuracy but also notably decreases computation time. This shows that this PL fitting method is suited for the design of learning rate schedulers that demand low computation overhead.

**Empirical analysis results.** We conduct an empirical analysis of `TempBalance` to discuss why it provides improvement. Our first analysis involves visualization to demonstrate how `TempBalance`

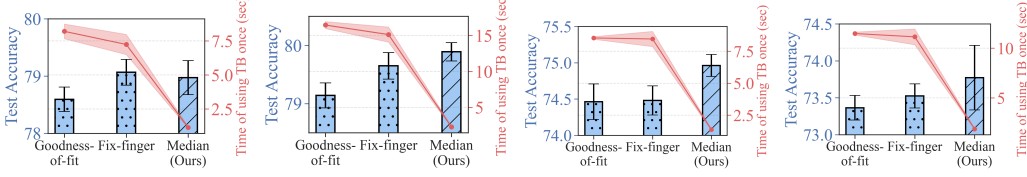

(a) ResNet18, CIFAR100  (b) ResNet34, CIFAR100  (c) VGG16, CIFAR100  (d) VGG19, CIFAR100

Figure 8: **(Varying PL fitting method to determine the $\lambda_{\text{min}}$).** Results of using different PL fitting methods. The blue bar plot and the left $y$-axis label denote the test accuracy (higher the better), and the red line plots and the right $y$-axis label denote the time in seconds of using `TempBalance` once (lower the better). Our design (`Median`) used in the proposed method achieves higher test accuracy and takes lower computation times compared to `Goodness-of-fit` and `Fix-finger`. The test accuracy is averaged over five random seeds and computation time is averaged over ten times.

effectively regularizes ESDs by scheduling the learning rate (see Appendix B). The second analysis strengthens the connections between `TempBalance` and HT structure, illustrating that the observed improvements are not due to indirectly addressing other training issues, such as gradient excursions [80] (see Appendix F).

**Corroborating results on other tasks.** We extend our evaluation of `TempBalance` to two additional tasks: object detection and language modeling, the details of which can be found in Appendix G. Across these tasks, `TempBalance` consistently outperforms the baseline `CAL` in terms of generalization.

## 5 Conclusion

Our extensive empirical evaluations demonstrate that `TempBalance` offers a straightforward yet effective layer-wise learning rate schedule. Our approach for balancing layer-wise temperature confirms the following: (i) HT-SR-motivated metric `PL_Alpha_Hill` helps layers achieve temperature balance during training, exhibits strong correlations with model quality, and yields improved performance during testing; (ii) temperature balancing is a novel and essential aspect of NN training, and HT-SR Theory provides a strong theoretical support for balancing temperatures; and (iii) layer-wise learning rate schedules are cheap and effective to apply, and it is useful to study these layer-wise learning rate schedules further. Our method provides insights into the study of layer-wise tuning approaches and load-temperature balancing in deep NN training, as it serves both as a layer-wise learning rate schedule and an effective regularization technique based on HT-SR Theory.

**Future directions, limitations, and societal impacts.** Our paper leaves many future directions to explore, of which we mention just a few.

- Can HT-SR metrics be extended to parameter-wise learning rate schedules, global learning rate schedules, or other hyperparameters? It would be of interest to observe how HT-SR can assist in acquiring a comprehensive set of hyperparameter tuning tools.
- Is it possible to accelerate the computation of ESDs and `PL_Alpha_Hill` to achieve a more adaptive learning rate scheduler? Currently, we calculate layer-wise `PL_Alpha_Hill` once per epoch, resulting in a minimal increase in computational cost. Consider the example of training ResNet18 for 200 epochs on CIFAR100. Calculating layer-wise `PL_Alpha_Hill` takes 1.14 seconds for each epoch, leading to 3.8 minutes in total. Training CIFAR100 on 1 Quadro RTX 6000 takes 59 minutes, and thus using `TB` increases 6% of training time. However, if we can significantly decrease the expense of computing ESDs, it might enable an optimizer that adjusts the learning rate every few gradient updates. A study on computation overhead is detailed in Appendix H.

Our research centers around developing a generic algorithm for optimizing NNs. Although `TempBalance` could be applied to learning models with adverse applications, we do not see any immediate negative societal impacts stemming from the algorithm itself. Indeed, we see a lot of societal value in using a practical, predictive, and quantitative theory, such as HT-SR Theory, as opposed to developing a method that relies on a theory that provides vacuous upper bounds and then relies on extremely expensive hyperparameter tuning to obtain good results.

**Acknowledgements.** WeightWatcher is a publicly available tool distributed under Apache License 2.0 with a copyright held by Calculation Consulting. Our conclusions do not necessarily reflect the position or the policy of our sponsors, and no official endorsement should be inferred.

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

# Appendix

## A  Heavy-tail phenomena in different DNN matrices are closely related

Recently, several papers have separately studied HT structures in different types of matrices, including the Hessian, the Fisher Information Matrix (FIM), and input/output covariance matrices [35, 81, 82]. The results confirm that when NNs are well-trained, various matrices have HT properties. Among these works, there are two major ways to characterize the HT spectrum, namely the HT-shaped ESDs (such as `PL_Alpha_Hill`), or HT-shaped decaying eigenvalues [33–35]. Our paper mainly uses the first way of characterizing the HT structure. On the other hand, the second way is to sort eigenvalues from largest to smallest and study the PL phenomena between the ordered eigenvalues and their index. Our experiments show fruitful connections between the PL phenomena manifested in different DNN matrices; if one matrix shows a PL spectrum, the other matrices often show something similar [35]. Thus, it is meaningful to ask why and how the PL phenomena in different prior works correlate.

This section first establishes the connections between input/output covariance matrices, the FIM and the Hessian in subsection A.1. We find that if one of these matrices shows the PL phenomenon, the other two matrices have a high chance to exhibit a similar PL phenomenon. Then, in subsection A.2, we derive the connection between our metric `PL_Alpha_Hill` and the PL exponent on decaying eigenvalues, showing a simple reciprocal relationship between these two.

### A.1  Connections between different matrices

Consider a NN $f_\theta : \mathbb{R}^d \to \mathbb{R}^C$, where $\theta \in \mathbb{R}^P$ is the vectorized weights, $d$ is the input dimension, and $C$ is the output dimension. When the NN is used for a classifying task, $C$ is also the number of classes. We denote the input data as $\{(x_i, y_i)\}_{i=1}^n$, where $x_i \in \mathbb{R}^d$, and the number of samples is $n$. We denote the loss function as $L(\theta) = \frac{1}{n} \sum_{i=1}^n l(y_i, f_\theta(x_i))$.

**Covariance matrices**. We denote the output covariance matrix as $\mathbb{E}[f_\theta(x) f_\theta^\top(x)]$, where the expectation is taken over the input distribution. We tend to consider the following empirical covariance matrix:

$$C(\theta) := \frac{1}{n} \sum_{i=1}^n f_\theta(x_i) f_\theta^\top(x_i) \in \mathbb{R}^{C \times C}. \tag{5}$$

**Fisher Information Matrices**. We denote the (output) FIM as

$$\mathbb{E}[\nabla_\theta f_\theta(x) \nabla_\theta f_\theta(x)^\top] = \sum_{k=1}^C \mathbb{E}[\nabla_\theta f_\theta^{(k)}(x) \nabla_\theta f_\theta^{(k)}(x)^\top], \tag{6}$$

where $f_\theta^{(k)}(x)$ is the $k$-th entry of the vector function $f(x)$. We also consider the empirical version of the FIM:

$$F(\theta) := \sum_{k=1}^C \frac{1}{n} \sum_{i=1}^n \nabla_\theta f_\theta^{(k)}(x_i) \nabla_\theta f_\theta^{(k)}(x_i)^\top \in \mathbb{R}^{P \times P}. \tag{7}$$

Note that (7) can be equally written as

$$F(\theta) := \frac{1}{n} \nabla_\theta \tilde{f}_\theta(x) \nabla_\theta \tilde{f}_\theta(x)^\top, \tag{8}$$

where $\nabla_\theta \tilde{f}_\theta(x)$ has the following form:

$$\begin{bmatrix} \frac{\partial f_\theta^{(1)}(x_1)}{\partial \theta_1} \cdots \frac{\partial f_\theta^{(1)}(x_n)}{\partial \theta_1} & \cdots & \frac{\partial f_\theta^{(C)}(x_1)}{\partial \theta_1} \cdots \frac{\partial f_\theta^{(C)}(x_n)}{\partial \theta_1} \\ \vdots & \ddots & \vdots \\ \frac{\partial f_\theta^{(1)}(x_1)}{\partial \theta_P} \cdots \frac{\partial f_\theta^{(1)}(x_n)}{\partial \theta_P} & \cdots & \frac{\partial f_\theta^{(C)}(x_1)}{\partial \theta_P} \cdots \frac{\partial f_\theta^{(C)}(x_n)}{\partial \theta_P} \end{bmatrix} \in \mathbb{R}^{P \times Cn}.$$

**Hessian Matrices**. We denote the Hessian as $\mathbb{E}\left[\frac{\partial^2 l(y, f_\theta(x))}{\partial \theta^2}\right]$, and we tend to consider the empirical Hessian Matrices:

$$H(\theta) := \frac{\partial^2 L(\theta)}{\partial \theta^2} \in R^{P \times P}, \tag{9}$$

where $L(\theta)$ is the empirical loss function $L(\theta) = \frac{1}{n}\sum_{i=1}^{n} l(y_i, f_\theta(x_i))$.

**Hessian and FIM are equivalent under certain conditions.** FIM can be defined in alternative ways different from (6). For instance, from classic statistical knowledge, we have the standard FIM (sFIM) in the following form:

$$sFIM := \mathbb{E}[\nabla_\theta \log P(y|x; \theta)\nabla_\theta \log P(y|x; \theta)^T], \tag{10}$$

where $P(y|x; \theta)$ represents the likelihood. After simple derivations, one can show that sFIM also has the following form [83, 84]:

$$sFIM = -\mathbb{E}\left[\frac{\partial^2 \log P(y|x; \theta)}{\partial \theta^2}\right]. \tag{11}$$

Therefore, when the loss function is defined as the negative log-likelihood, the sFIM in (11) is equivalent to Hessian defined in (9).

**Why is the FIM defined in** (6) **equivalent to** (10). Back to deep learning, the FIM is often defined as (6). It is thus meaningful to derive the equivalence between these two forms. Suppose $P(y|x; \theta)$ here means the conditional probability distribution of output $y$ given input data $x$. If $P(y|x; \theta)$ is assumed to take the following form:

$$P(y|x; \theta) = \frac{1}{\sqrt{2\pi}} \exp\left(-\frac{1}{2}\|y - f_\theta(x)\|^2\right), \tag{12}$$

then the MSE estimator $\min_\theta \frac{1}{2}\|y - f_\theta(x)\|^2$ is equivalent to the maximum likelihood estimation of $P(y|x; \theta)$. Then, plugging (12) into (10), we have:

$$sFIM_{mse} = \mathbb{E}[\|y - f_\theta(x)\|^2 \nabla_\theta f_\theta(x)\nabla_\theta f_\theta(x)^T]. \tag{13}$$

We now expand $sFIM_{mse}$ by the definition of expectation, and we have the following [81]:

$$sFIM_{mse} = \int_{\mathbb{R}} \int_{\mathbb{R}} \|y - f_\theta(x)\|^2 \nabla_\theta f_\theta(x)\nabla_\theta f_\theta(x)^T p(x, y; \theta)dydx \tag{14}$$

$$= \int_{\mathbb{R}} \int_{\mathbb{R}} \|y - f_\theta(x)\|^2 \nabla_\theta f_\theta(x)\nabla_\theta f_\theta(x)^T P(y|x; \theta)q(x)dydx \tag{15}$$

$$= \int_{\mathbb{R}} \left[\int_{\mathbb{R}} \frac{1}{\sqrt{2\pi}}\|y - f_\theta(x)\|^2 \exp\left(-\frac{1}{2}\|y - f_\theta(x)\|^2\right)dy\right] \nabla_\theta f_\theta(x)\nabla_\theta f_\theta(x)^T q(x)dx \tag{16}$$

$$= \int_{\mathbb{R}} \nabla_\theta f_\theta(x)\nabla_\theta f_\theta(x)^T q(x)dx \tag{17}$$

$$= \mathbb{E}[\nabla_\theta f_\theta(x)\nabla_\theta f_\theta(x)^T], \tag{18}$$

where (14) follows from the definition of expectation, $q(x)$ is input distribution, and (17) holds because the integral of $y$ in the brackets [] equals 1 due to the property of Gamma function $\Gamma(\cdot)$.

Therefore, from (18), we find that $sFIM_{mse}$ is just equal to $FIM$, defined in (6). Also, plugging (12) into $\mathbb{E}\left[\frac{\partial^2 \log P(y|x; \theta)}{\partial \theta^2}\right]$ and taking the loss function $L(\theta)$ as the mean-square loss, we will again find that $\mathbb{E}\left[\frac{\partial^2 \log P(y|x; \theta)}{\partial \theta^2}\right]$ is equal to $H(\theta)$. Therefore, jointly considering (11), we can see that FIM is equal to the Hessian $H(\theta)$.

**PL in the covariance matrix and PL in Hessian are tightly correlated.** Next, we consider the relationship between the covariance matrix and the Hessian. Suppose the NN function $f_\theta$ is a Lipchitz function [85]. Then, it can be seen that the covariance matrix (5) may be controlled and estimated by FIM defined in (6), which is equivalent to being controlled by Hessian.

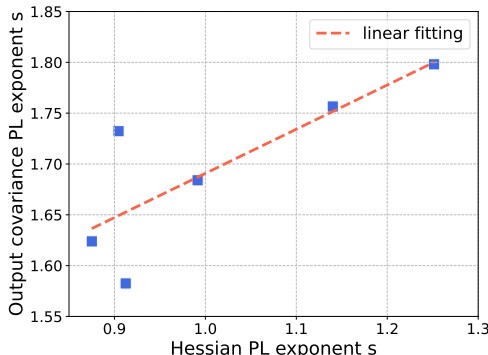

Figure 9: We train a MLP for 50 epochs and fit PL exponent $s$ for both the output covariance and the Hessian. For models trained with epochs {1, 10, 20, 30, 40, 50}, we see their PL exponents $s$ show a strong correlation.

Although deriving an exact equivalent between these two can be hard, we numerically show that the PL in one matrix informs the PL in the other. To visualize their relationship in the presence of PL, we train a simple MLP on MNIST [86] with one hidden layer and 2000 neurons for 50 epochs. We leverage the spectral regularization from Nassar et al. [34] to make the output covariance matrix exhibit a PL spectrum. Meanwhile, we calculate the top eigenvalues of the covariance and the Hessian [87], fit the PL exponent $s$ for each matrix, and compare the PL exponents against each other. More specifically, we take trained NNs from epochs {1, 10, 20, 30, 40, 50} and plot the Hessian PL exponent $s$ versus the output covariance PL exponent $s$. From the results shown in Figure 9, we can see that their PL exponent $s$ shows a strong correlation, which supports our claim that the PL phenomena in one matrix can inform the other.

**Connections to the NTK matrix.** Interestingly, if we ignore the constant in (8) and switch the two matrices multiplied together, we obtain $\nabla_\theta \tilde{f}_\theta(x)^T \nabla_\theta \tilde{f}_\theta(x)$. This matrix is equal to Neural Tangent Kernel(NTK) [88], which is a kernel used to approximate the deep NN when NN's width is infinite. We thus conjecture that NTK should show PL when the NN is well trained [89]. Indeed, Karakida et al. [81] and Karakida et al. [82] study the eigenvalues of NTK, showing a PL trend. Some other work on stochastic gradient [90] claim that the so-called "stochastic gradient matrix" (which is similar to the NTK matrix) shows a PL spectrum as well, which matches our expectations. Also, Lewkowycz et al. [91], Dyer and Gur-Ari [92] show that the eigenvalues of NTK are similar to those in the Hessian, which again meets our expectation because the Hessian tends to be PL when NNs are well-trained [35].

In summary, this section investigates different "important matrices" and shows that they are tightly correlated to each other in terms of the PL trends: if one matrix shows a PL spectrum, there is a high chance that the other ones show something similar.

### A.2 Connections between PL in ESD and PL in decaying eigenvalues

Next, we derive the connection between our `PL_Alpha_Hill` metric and the exponent of PL distribution on decaying eigenvalues. Take the covariance matrix (5) as an instance. According to Nassar et al. [34], the HT phenomenon in the output covariance matrix is similar to the layer-wise covariance matrices. Thus, without the loss of generality, we can consider the case when there is only one layer in the NN. We assume the weight matrix $L$ is in $\mathbf{R}^{N \times Q}$. According to prior works, when $L$ is well-trained, the ESD follows a PL distribution:

$$p(\lambda) = \frac{1}{H}\lambda^{-\alpha}, \quad \lambda_{\min} < \lambda < \lambda_{\max}. \tag{19}$$

Here, $H$ is a normalizing constant, and $\alpha$ is the PL exponent.

Another way to characterize the PL phenomenon is to consider eigenvalues directly following a PL series. For example, Xie et al. [35] show that the decaying eigenvalues follows the following PL

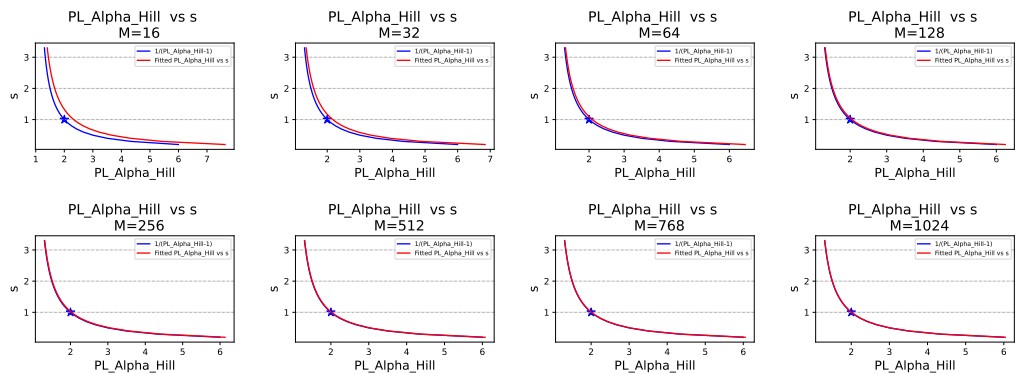

Figure 10: We show the connection between `PL_Alpha_Hill` and the PL exponent of the decaying eigenvalues (denoted as $s$) satisfy $s = \frac{1}{\texttt{PL\_Alpha\_Hill}-1}$. Results are shown for different matrix size $Q$. In particular, we see that `PL_Alpha_Hill` $= 2$ [19] is equivalent to $s = 1$ [33] in the linear case.

series:

$$\lambda_k = \lambda_1 k^{-s}, k = 1, 2, \cdots, Q, \tag{20}$$

where $\lambda_1$ is the same as $\lambda_{\max}$ used in the main paper.

Now, we will analytically and empirically show that these two ways of characterizing PL are strongly related. Furthermore, the two PL coefficients satisfy $s = \frac{1}{\alpha-1}$.

**An analytical way to show that $s = \frac{1}{\alpha-1}$.** The derivation is actually quite simple. Consider the case that $\lambda_k = \lambda_1 k^{-s}$ (i.e., (20) holds), and suppose $\Lambda$ is a random variable distributed according to the empirical distribution from these eigenvalues $\lambda_k = \lambda_1 k^{-s}$. Now, from (20), we can see that the distribution function takes the following form:

$$\mathbb{P}(\Lambda > \lambda_1 k^{-s}) = \frac{k}{Q}. \tag{21}$$

By changing variables $\lambda_1 k^{-s} = \lambda$, we get the cumulative distribution function of $\Lambda$:

$$\mathbb{P}(\Lambda > \lambda) \sim \lambda^{-\frac{1}{s}}. \tag{22}$$

After that, we take the derivative with respect to $\lambda$, and we get the ESD:

$$p(\lambda) \sim \lambda^{-(\frac{1}{s}+1)}. \tag{23}$$

In other words, we have $\lambda^{-(\frac{1}{s}+1)} = \lambda^{-\alpha}$, which means $s = \frac{1}{\alpha-1}$.

**An empirical way to show that $s = \frac{1}{\alpha-1}$.** We consider matrices of size $Q \times Q$, where we choose $Q$ in $\{16, 32, 64, 128, 256, 512, 768, 1024\}$, and we assign the parameters such that the decaying eigenvalues obey the formula $\lambda_1 k^{-s}$, for $s$ in $\{0.2, 0.3, 0.4, \cdots, 3.2\}$. Then, we fit the ESD and get our estimate `PL_Alpha_Hill`. We plot the relationship between `PL_Alpha_Hill` and $s$ in Figure10. From Figure 10, we find that the connection between `PL_Alpha_Hill` and $s$ shows a good fit with the formula $s = \frac{1}{\alpha-1}$. With increasing matrix size $Q$, the fitting becomes increasingly accurate.

**When $s = \frac{1}{\alpha-1}$, $s = 1$ corresponds to $\alpha = 2$.** Some prior works Nassar et al. [34], Xie et al. [35], Bartlett et al. [93] measure the HT phenomena from the perspective of decaying eigenvalues with PL exponent $s$, and they show either theoretically or empirically that $s = 1$ is the *optimal* exponent. Now that we have $s = \frac{1}{\alpha-1}$ in the linear case, and from the theory of NTK[88], the infinite wide NN is approximated as a linear model, we tend to believe that $\alpha = 2$ satisfies a similar property. Indeed, one of the main contributions of Martin and Mahoney [19] is to establish different HT families of ESDs, and $\alpha = 2$ is believed to be the boundary between "moderately HT" and "very HT," corresponding to the best models. Martin and Mahoney [19] further argue that the optimal exponent for `PL_Alpha` is in the range [2,4]. Combining the perspective from Nassar et al. [34], Xie et al. [35], Bartlett et al. [93] and those from Martin and Mahoney [19], it is reasonable to believe that the optimal exponent for `PL_Alpha` is around 2. When `PL_Alpha` is much higher or lower than 2,

the NN probably has some issue in training. Although we argued in the main paper that the absolute numerical value of `PL_Alpha` is unimportant in implementing our `TempBalance` algorithm, it is, however, helpful to have an "optimal" `PL_Alpha` value to test if our algorithm actually works in controlling the ESDs. We will show visualization results in Appendix B that `TempBalance` leads to a better distribution of our estimated `PL_Alpha_Hill`.

In summary, this section explores two distinct methods for determining PL fit. We demonstrate that, although these two methods yield numerically distinct PL exponents, they essentially capture the same underlying phenomenon. Moreover, it is noteworthy that the "optimal" values of the PL exponents reported in various papers are consistent with one another [19, 34, 35, 93].

## B  Visualization results: how does `TempBalance` control ESDs

We demonstrate that the proposed method, `TempBalance`, effectively controls the shape of ESDs, resulting in a more favorable distribution of `PL_Alpha_Hill` among the layers of NNs compared to the baseline method `CAL`. This observation elucidates the superior performance of `TempBalance` over `CAL` in our main experiment, as presented in Section 4.2.

We evaluate the models reported in the main paper. For each individual NN, we compute and aggregate `PL_Alpha_Hill` values across all layers, excluding the first and last layers that have an extremely small number of eigenvalues and thus cause inaccurate `PL_Alpha_Hill` estimation. We aggregate the `PL_Alpha_Hill` values from five models trained using different random seeds for each method. Figure 11 shows the distribution of `PL_Alpha_Hill` of `TempBalance` and the baseline `CAL`. Comparing `TempBalance` with `CAL`, we see that `TempBalance` consistently yields a more concentrated distribution. Furthermore, `TempBalance` causes the median and mean of the distribution to approach 2 (shown in each subplot respectively as the middle vertical line and the red star). The value 2 represents the theoretically optimal `PL_Alpha_Hill` value, as we have justified in Appendix A.

Next, in Figure 12, we group the models into different subgroups based on their architectures and/or datasets, aggregating the `PL_Alpha_Hill` values and comparing the distributions of the two methods `TempBalance` and `CAL`. Once again, we observe that `TempBalance` results in a more concentrated distribution, with a larger number of samples (layers) having `PL_Alpha_Hill` values closer to 2.

We provide visualization to demonstrate how the learning rates are distributed over layers during the training. In Figure 13, we report the learning rate and `PL_Alpha_Hill` every epoch throughout the 200-epoch training duration. The key observation includes the following.

1. **How does the learning rate vary across layers?** We observed a correlation between the layer-wise learning rate and the layer-wise `PL_Alpha_Hill` distribution: layers with larger `PL_Alpha_Hill` are allocated larger learning rates, whereas those with smaller `PL_Alpha_Hill` receive smaller learning rates.

2. **How does the layer-wise learning rate evolve during training?** The variations in layer-wise learning rates closely reflect shifts in the layer-wise `PL_Alpha_Hill` distribution. Initially, the `PL_Alpha_Hill` distributes uniformly across layers but eventually converge to a layer-wise pattern where earlier layers have smaller `PL_Alpha_Hill` and later layers have larger ones.

We present visualizations of how `PL_Alpha_Hill` and learning rate evolve through training. In Figure 14, we show `PL_Alpha_Hill` and learning rate of two layers within the same ResNet18 during the training process. The two layers are $\text{layer}1.0.\text{conv}2$ (index=1) and $\text{layer}4.0.\text{conv}2$ (index=15). From Figure 14b and 14d, we can see that with the baseline `CAL` scheduler (blue curves), the earlier layer (index=1) achieves a smaller `PL_Alpha_Hill` value compared to the larger `PL_Alpha_Hill` value of the later layer (index=15). In contrast, `TempBalance` (orange curves) narrows this gap, indicating our approach balances the undertraining/overtraining levels (as signified by `PL_Alpha_Hill`) of different layers. This balancing effect is further corroborated by Figures 11 and 12 , where our method consistently refines the layer-wise `PL_Alpha_Hill` distribution. Regarding the learning rate plots in Figure 14a and 14c, `TempBalance` allocates a lower learning rate for earlier layers and a higher one for later layers than the baseline does. This leads to a more balanced `PL_Alpha_Hill` distribution between layers as mentioned above. Additionally, we noted

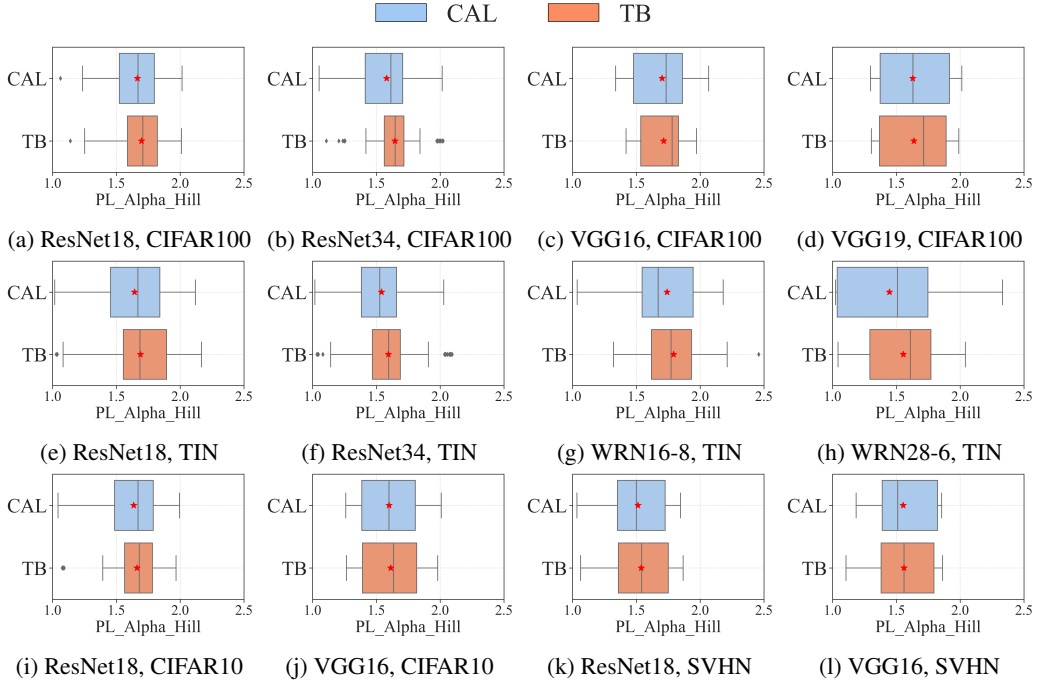

Figure 11: Comparing the distribution of `PL_Alpha_Hill` of NNs trained by our method `TempBalance` (TB) and `CAL`. The mean of each distribution is indicated by a red star marker. Each distribution aggregates the `PL_Alpha_Hill` values from models trained using five different random seeds. Across all experiments, our method `TempBalance` consistently yields a more concentrated distribution, resulting in the mean and median approaching the theoretically optimal `PL_Alpha_Hill` value of 2, as supported in Appendix A.

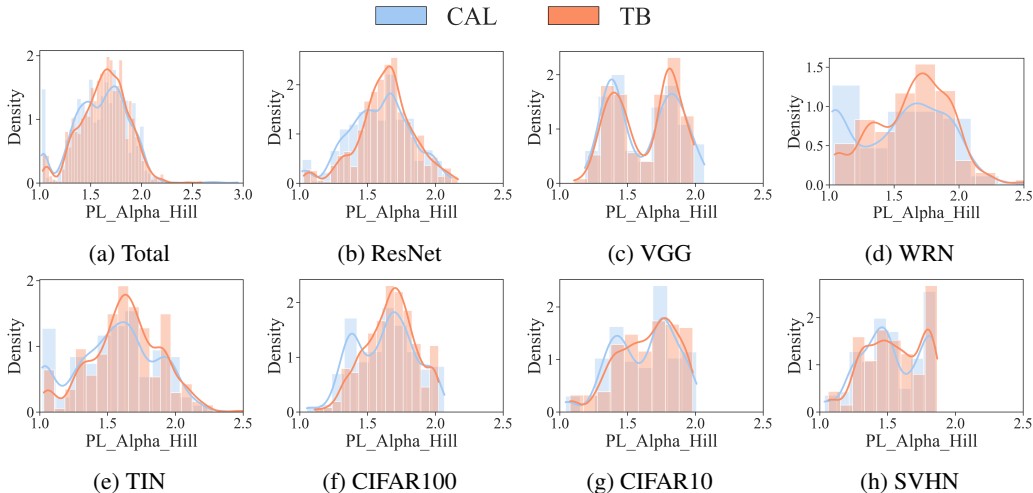

Figure 12: Comparing our method `TempBalance` (TB) to `CAL` in terms of the distribution of `PL_Alpha_Hill` of aggregating NNs into different architectures and datasets. Each distribution aggregates the `PL_Alpha_Hill` of models trained with five random seeds. Across all subgroups, our method `TempBalance` consistently exhibits a more concentrated distribution, accompanied by a higher number of layers approaching a `PL_Alpha_Hill` value close to 2. This value of 2 corresponds to the theoretically optimal `PL_Alpha_Hill` value, as justified in Appendix A.

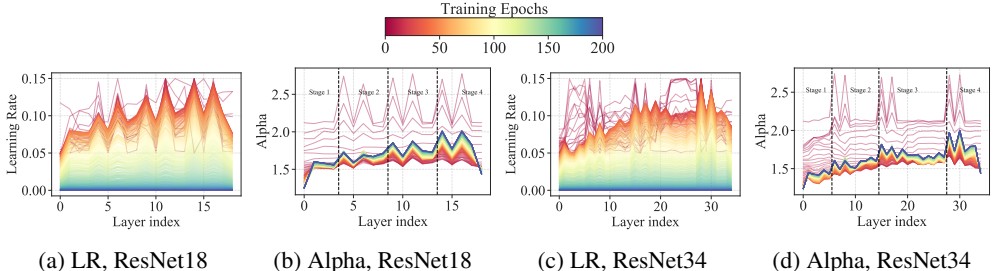

| (a) LR, ResNet18 | (b) Alpha, ResNet18 | (c) LR, ResNet34 | (d) Alpha, ResNet34 |

Figure 13: **(Visualization of layer-wise learning rate (LR) and `PL_Alpha_Hill` (Alpha) over training)**. (a-b) The layer-wise LR and `PL_Alpha_Hill` of ResNet18 over training. (c-d) The layer-wise LR and `PL_Alpha_Hill` of ResNet34 over training.

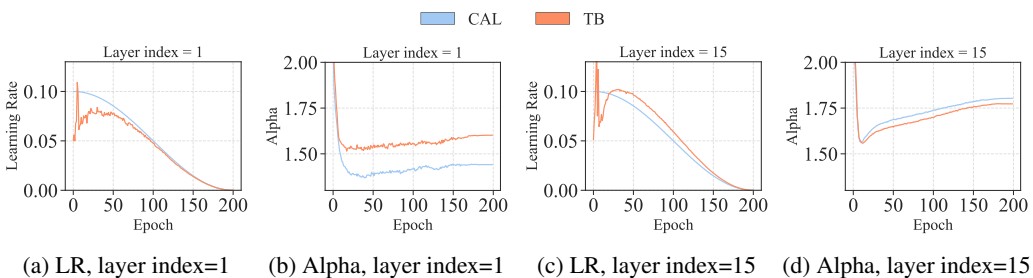

| (a) LR, layer index=1 | (b) Alpha, layer index=1 | (c) LR, layer index=15 | (d) Alpha, layer index=15 |

Figure 14: **(Visualization of learning rate (LR) and `PL_Alpha_Hill` (Alpha) of two layers during training)** (a-b) LR and `PL_Alpha_Hill` of one layer with index = 1 in ResNet18. (c-d) LR and `PL_Alpha_Hill` of one layer with index = 15 in ResNet18. The ResNet18 is trained on CIFAR100.

instability in the learning rate curves during early training phases, while smoother transitions emerge in later phases.

## C   Ablation studies

We provide additional ablation studies on the choices of learning rate assignment function, assignment hyperparameters.

**Varying LR assignment function.**   For `TempBalance`, we selected the linear interpolation (Equation 2) for learning rate assignment function $f_t$, based on its superior performance in our ablation study.

We evaluated three alternative learning rate assignment functions: Square root (Sqrt), Log2, and Step:

- Sqrt : $f_t(i) = \eta_t \dfrac{\sqrt{\alpha_t^i}}{\frac{1}{L}\sum_{j=1}^{L}\sqrt{\alpha_t^j}}$,
- Log2: $f_t(i) = \eta_t \dfrac{log(\alpha_t^i)}{\frac{1}{L}\sum_{j=1}^{L} log(\alpha_t^j)}$,
- Step: For layer $i$ with $k$-th minimum `PL_Alpha_Hill` among all the layers,

$$f_t(i) = \eta_t(s_1 + (k-1)\frac{s_2 - s_1}{L - 1})$$

Here, $\eta_t$ denotes the base global learning rate at epoch $t$, $(s_1, s_2)$ represents the minimum and maximum learning rate scaling ratios relative to $\eta_t$, $\alpha_t^i$ is the `PL_Alpha_Hill` estimate of the layer $i$ at epoch $t$, and $L$ is the total number of model layers. All these notations are consistently used in the main paper.

As depicted in Figure 15, `TempBalance` (TB), with the current assignment function, surpasses the other designs when tested on VGG and ResNet architectures on CIFAR100. All hyperparameters are consistent with the main paper. Each experiment was conducted with five random seeds.

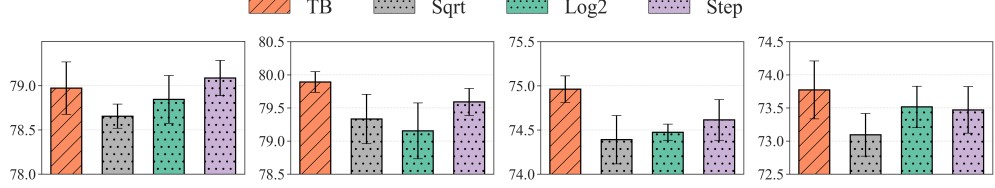

(a) ResNet18, CIFAR100   (b) ResNet34, CIFAR100   (c) VGG16, CIFAR100   (d) VGG19, CIFAR100

Figure 15: (**Different designs for learning rate assignment function**.) Results of using different learning rate assignment functions on different architectures and CIFAR-100. Our design in the main paper `TempBalance` (TB) outperforms others. Reporting mean/std over five random seeds.

**Varying LR assignment function hyperparameters.** We provide additional results of a hyperparameter study on $(s_1, s_2)$, in which we consider five different settings for $(s_1, s_2)$: $[(0.5, 1.5), (0.6, 1.4), (0.7, 1.3), (0.8, 1.2), (0.9, 1.1)]$. We run tasks on CIFAR100 with four VGG and ResNet architectures, each with five random seeds. Our results in Figure 16 show that a larger learning rate scaling range $(0.5, 1.5)$ performs best. This hyperparameter setting is the default setting used in our paper. All hyperparameters are consistent with those described in the main paper.

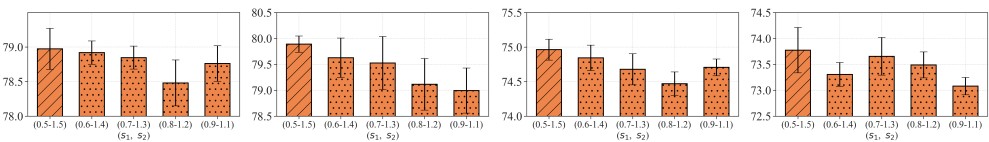

(a) ResNet18, CIFAR100   (b) ResNet34, CIFAR100   (c) VGG16, CIFAR100   (d) VGG19, CIFAR100

Figure 16: (**Hyperparameter study on $(s_1, s_2)$**). Search for hyperparameters $(s_1, s_2)$ with different architectures on CIFAR100. The current hyperparameter choice $(0.5, 1.5)$ used in the paper performs best among all the cases. Reporting mean/std over five random seeds.

# D   Hyperparameter settings for reproducing our results

We report all hyperparameters, random seeds and all numerical values of experimental results shown in the main paper (in Section 4).

First, we report the common hyperparameters shared by all the experiments: the default optimizer is SGD, trained with batch size 128, number of training epochs 200, weight decay 5e-4, and momentum 0.9. The default HT-SR metric used in `TempBalance` is `PL_Alpha_Hill`. For each experimental setting, we use five random seeds, which are always 43, 37, 13, 51, 71, and we report the mean and standard deviation of the test accuracy across these seeds.

First, Table 1 reports the details of experiments shown in Figure 3. We carefully tune the initial learning rate $\eta_0$ and $\lambda_{sr}$ for the two baseline methods `CAL` and `SNR`. Then, Table 2 reports the detailed hyperparameter settings of the experiments shown in Figure 4. We again carefully tune the hyperparameters of various baseline optimizers and schedulers, as specified in their papers. Finally, Table 3, Table 4, Table 5 and Tabel 6 respectively report the details of the experiments shown in Figure 5, Figure 6, Figure 7 and Figure 8.

Table 1: Parameter settings of the experiment reported in Section 4.2 Figure 3. The hyperparameter in bold is the best hyperparameter selection reported in the main paper. The five random seeds for each setting are {43, 37, 13, 51, 71}, and the means and standard deviations of the test accuracy among the five seeds are reported.

| Index | Dataset | Model | Method | Initial learning rate $\eta_0$ | $\lambda_{sr}$ | Test Acc (best hyperparam.) | scaling ratio $(s_1, s_2)$ |
|---|---|---|---|---|---|---|---|
| 0 | | ResNet18 | CAL | 0.05, **0.1**, 0.15 | - | $78.31 \pm 0.05$ | - |
| 1 | | ResNet18 | SNR | 0.1 | 0.001, 0.005, **0.01**, 0.015 | $78.65 \pm 0.29$ | - |
| 2 | | ResNet18 | TB | 0.1 | - | $78.97 \pm 0.29$ | (0.5, 1.5) |
| 3 | | ResNet18 | TB + SNR | 0.1 | 0.001 | $79.06 \pm 0.32$ | (0.6, 1.4) |
| 4 | | ResNet34 | CAL | **0.05**, 0.1, 0.15 | - | $78.98 \pm 0.14$ | - |
| 5 | | ResNet34 | SNR | 0.1 | 0.001, 0.005, 0.01, **0.015** | $79.97 \pm 0.21$ | - |
| 6 | | ResNet34 | TB | 0.1 | - | $79.89 \pm 0.15$ | (0.5, 1.5) |
| 7 | CIFAR100 | ResNet34 | TB + SNR | 0.1 | 0.005 | $80.09 \pm 0.35$ | (0.6, 1.4) |
| 8 | | VGG16 | CAL | 0.025, **0.05**, 0.1 | - | $74.59 \pm 0.23$ | - |
| 9 | | VGG16 | SNR | 0.05 | 0.001, **0.005**, 0.01, 0.015 | $74.80 \pm 0.28$ | - |
| 10 | | VGG16 | TB | 0.05 | - | $74.96 \pm 0.15$ | (0.5, 1.5) |
| 11 | | VGG16 | TB + SNR | 0.05 | 0.005 | $75.52 \pm 0.46$ | (0.6, 1.4) |
| 12 | | VGG19 | CAL | 0.025, **0.05**, 0.1 | - | $73.26 \pm 0.37$ | - |
| 13 | | VGG19 | SNR | 0.05 | 0.001, 0.005, **0.01**, 0.015 | $74.37 \pm 0.16$ | - |
| 14 | | VGG19 | TB | 0.05 | - | $73.77 \pm 0.43$ | (0.5, 1.5) |
| 15 | | VGG19 | TB + SNR | 0.05 | 0.01 | $74.74 \pm 0.10$ | (0.5, 1.5) |
| 16 | | ResNet18 | CAL | 0.05, **0.1**, 0.15 | - | $66.25 \pm 0.17$ | - |
| 17 | | ResNet18 | SNR | 0.1 | 0.001, 0.005, **0.01**, 0.015 | $66.20 \pm 0.22$ | - |
| 18 | | ResNet18 | TB | 0.1 | - | $66.77 \pm 0.25$ | (0.6, 1.4) |
| 19 | | ResNet18 | TB + SNR | 0.1 | 0.001 | $66.86 \pm 0.22$ | (0.6, 1.4) |
| 20 | | ResNet34 | CAL | 0.05, **0.1**, 0.15 | - | $68.19 \pm 0.16$ | - |
| 21 | | ResNet34 | SNR | 0.1 | 0.001, 0.005, **0.01**, 0.015 | $68.69 \pm 0.13$ | - |
| 22 | | ResNet34 | TB | 0.1 | - | $69.12 \pm 0.16$ | (0.6, 1.4) |
| 23 | | ResNet34 | TB + SNR | 0.1 | 0.001 | $69.27 \pm 0.21$ | (0.6, 1.4) |
| 24 | TinyImageNet | WRN16-8 | CAL | 0.05, **0.1**, 0.15 | - | $63.67 \pm 0.09$ | - |
| 25 | | WRN16-8 | SNR | 0.1 | 0.00005, **0.0001**, 0.001 | $63.98 \pm 0.23$ | - |
| 26 | | WRN16-8 | TB | 0.1 | - | $64.09 \pm 0.17$ | (0.6, 1.4) |
| 27 | | WRN16-8 | TB + SNR | 0.1 | 0.0001 | $64.08 \pm 0.07$ | (0.6, 1.4) |
| 28 | | WRN28-6 | CAL | **0.05**, 0.1, 0.15 | - | $65.88 \pm 0.20$ | - |
| 29 | | WRN28-6 | SNR | 0.1 | 0.00005, **0.0001**, 0.001 | $66.09 \pm 0.25$ | - |
| 30 | | WRN28-6 | TB | 0.1 | - | $66.58 \pm 0.23$ | (0.6, 1.4) |
| 31 | | WRN28-6 | TB + SNR | 0.1 | 0.0001 | $66.79 \pm 0.25$ | (0.6, 1.4) |
| 32 | | ResNet18 | CAL | 0.05, **0.1**, 0.15 | - | $95.53 \pm 0.12$ | - |
| 33 | | ResNet18 | SNR | 0.1 | **0.001**, 0.005, 0.01, 0.015 | $95.57 \pm 0.06$ | - |
| 34 | | ResNet18 | TB | 0.1 | - | $95.63 \pm 0.08$ | (0.5, 1.5) |
| 35 | CIFAR10 | ResNet18 | TB + SNR | 0.1 | 0.001 | $95.66 \pm 0.09$ | (0.6, 1.4) |
| 36 | | VGG16 | CAL | 0.025, 0.05, **0.1** | - | $93.98 \pm 0.12$ | - |
| 37 | | VGG16 | SNR | 0.05 | 0.001, **0.005**, 0.01, 0.015 | $94.04 \pm 0.07$ | - |
| 38 | | VGG16 | TB | 0.05 | - | $94.14 \pm 0.06$ | (0.5, 1.5) |
| 39 | | VGG16 | TB + SNR | 0.05 | 0.005 | $94.26 \pm 0.10$ | (0.6, 1.4) |
| 40 | | ResNet18 | CAL | 0.05, **0.1**, 0.15 | - | $96.59 \pm 0.08$ | - |
| 41 | | ResNet18 | SNR | 0.1 | 0.001, **0.005**, 0.015, 0.01 | $96.65 \pm 0.12$ | - |
| 42 | | ResNet18 | TB | 0.1 | - | $96.63 \pm 0.06$ | (0.5, 1.5) |
| 43 | | ResNet18 | TB + SNR | 0.1 | 0.01 | $96.67 \pm 0.09$ | (0.6, 1.4) |
| 44 | SVHN | VGG16 | CAL | 0.025, **0.05**, 0.1 | - | $96.28 \pm 0.04$ | - |
| 45 | | VGG16 | SNR | 0.05 | 0.001, 0.005, **0.015**, 0.01 | $96.32 \pm 0.07$ | - |
| 46 | | VGG16 | TB | 0.05 | - | $96.33 \pm 0.06$ | (0.5, 1.5) |
| 47 | | VGG16 | TB + SNR | 0.05 | 0.005 | $96.40 \pm 0.08$ | (0.6, 1.4) |

Table 2: Parameter settings of the experiment reported in Section 4.2 Figure 4. The hyperparameter in bold is the best hyperparameter selection reported in the main paper. The five random seeds for each setting are {43, 37, 13, 51, 71}, and the means and standard deviations of the test accuracy among the five seeds are reported.

| Index | Dataset | Model | Method | Initial learning rate $\eta_0$ | SGDR $(T_0, T_{mut})$ | Lookahead $k$ | Lookahead $\alpha$ | Test Acc (best hyperparams.) | scaling ratio $(s_1, s_2)$ |
|---|---|---|---|---|---|---|---|---|---|
| 0 | | ResNet18 | CAL | 0.05, **0.1**, 0.15 | - | - | - | $78.31 \pm 0.05$ | - |
| 1 | | ResNet18 | SGDR | 0.05, **0.1**, 0.15 | **(100,1)**, (10, 2),(1, 2) | - | - | $77.69 \pm 0.20$ | - |
| 2 | | ResNet18 | LARS | 26, **28**, 30, 32, 34 | - | - | - | $78.44 \pm 0.12$ | - |
| 3 | | ResNet18 | Lookahead | 0.05, **0.1**, 0.15 | - | **10**, 5 | **0.8**, 0.5 | $78.46 \pm 0.18$ | - |
| 4 | | ResNet18 | SGDP | 0.01, 0.05, **0.1**, 0.15, 0.2 | - | - | - | $78.74 \pm 0.11$ | - |
| 5 | | ResNet18 | TB | 0.05, **0.1**, 0.15 | - | - | - | $78.97 \pm 0.29$ | (0.5, 1.5) |
| 6 | | ResNet18 | TB + SGDP | 0.05, **0.1**, 0.15 | - | - | - | $79.13 \pm 0.15$ | (0.5, 1.5) |
| 7 | CIFAR100 | ResNet34 | CAL | **0.05**, 0.1, 0.15 | - | - | - | $78.98 \pm 0.14$ | - |
| 8 | | ResNet34 | SGDR | **0.05**, 0.1, 0.15 | **(100,1)**, (10, 2), (1, 2) | - | - | $78.61 \pm 0.20$ | - |
| 9 | | ResNet34 | LARS | 26, 28, 30, **32**, 34 | - | - | - | $78.94 \pm 0.19$ | - |
| 10 | | ResNet34 | Lookahead | 0.05, 0.1, **0.15** | - | **10**, 5 | **0.8**, 0.5 | $79.19 \pm 0.12$ | - |
| 11 | | ResNet34 | SGDP | 0.01, 0.05, **0.1**, 0.15, 0.2 | - | - | - | $79.34 \pm 0.21$ | - |
| 12 | | ResNet34 | TB | 0.05, **0.1**, 0.15 | - | - | - | $79.89 \pm 0.15$ | (0.5, 1.5) |
| 13 | | ResNet34 | TB + SGDP | 0.05, **0.1**, 0.15 | - | - | - | $79.94 \pm 0.30$ | (0.5, 1.5) |

Table 3: Parameter settings of the experiment reported in Section 4.3 Figure 5. The five random seeds for each setting are {43, 37, 13, 51, 71}, and the means and standard deviations of the test accuracy among the five seeds are reported.

| Index | Dataset | Model | Method | Initial learning rate $\eta_0$ | Test Acc | scaling ratio $(s_1, s_2)$ |
|---|---|---|---|---|---|---|
| 0 | | ResNet18 | CAL | 0.05, 0.1, 0.15 | $78.08 \pm 0.19, 78.31 \pm 0.05, 77.72 \pm 0.44$ | - |
| 1 | | ResNet18 | TB | 0.05, 0.1, 0.15 | $78.48 \pm 0.27, 78.97 \pm 0.29, 78.69 \pm 0.11$ | (0.5, 1.5) |
| 2 | | ResNet34 | CAL | 0.05, 0.1, 0.15 | $78.98 \pm 0.14, 78.89 \pm 0.24, 78.51 \pm 0.34$ | - |
| 3 | CIFAR100 | ResNet34 | TB | 0.05, 0.1, 0.15 | $79.36 \pm 0.18, 79.89 \pm 0.15, 79.09 \pm 0.64$ | (0.5, 1.5) |
| 4 | | VGG16 | CAL | 0.025, 0.05, 0.1 | $73.96 \pm 0.27, 74.59 \pm 0.23, 74.46 \pm 0.12$ | - |
| 5 | | VGG16 | TB | 0.025, 0.05, 0.1 | $74.40 \pm 0.31, 74.96 \pm 0.15, 74.94 \pm 0.16$ | (0.5, 1.5) |
| 6 | | VGG19 | CAL | 0.025, 0.05, 0.1 | $72.57 \pm 0.45, 73.26 \pm 0.37, 72.98 \pm 0.16$ | - |
| 7 | | VGG19 | TB | 0.025, 0.05, 0.1 | $73.47 \pm 0.16, 73.77 \pm 0.43, 73.40 \pm 0.38$ | (0.5, 1.5) |

Table 4: Parameter settings of the experiment reported in Section 4.3 Figure 6. The five random seeds for each setting are {43, 37, 13, 51, 71}, and the means and standard deviations of the test accuracy among the five seeds are reported.

| Index | Dataset | Model | Method | Initial learning rate $\eta_0$ | Width | Test Acc | scaling ratio $(s_1, s_2)$ |
|---|---|---|---|---|---|---|---|
| 0 | | ResNet18 | CAL | 0.1 | 256, 512, 768 | $75.05 \pm 0.26, 78.31 \pm 0.05, 79.44 \pm 0.26$ | - |
| 1 | | ResNet18 | TB | 0.1 | 256, 512, 768 | $75.63 \pm 0.12, 78.97 \pm 0.29, 80.47 \pm 0.18$ | (0.5, 1.5) |
| 2 | | ResNet34 | CAL | 0.1 | 256, 512, 768 | $76.79 \pm 0.34, 78.89 \pm 0.24, 79.94 \pm 0.31$ | - |
| 3 | CIFAR100 | ResNet34 | TB | 0.1 | 256, 512, 768 | $77.25 \pm 0.14, 79.89 \pm 0.15, 80.23 \pm 0.53$ | (0.5, 1.5) |
| 4 | | VGG16 | CAL | 0.05 | 256, 512, 768 | $71.04 \pm 0.14, 74.59 \pm 0.23, 75.53 \pm 0.32$ | - |
| 5 | | VGG16 | TB | 0.05 | 256, 512, 768 | $71.26 \pm 0.26, 74.96 \pm 0.15, 76.19 \pm 0.14$ | (0.5, 1.5) |
| 6 | | VGG19 | CAL | 0.05 | 256, 512, 768 | $69.58 \pm 0.39, 73.26 \pm 0.37, 74.39 \pm 0.33$ | - |
| 7 | | VGG19 | TB | 0.05 | 256, 512, 768 | $69.96 \pm 0.25, 73.77 \pm 0.43, 74.80 \pm 0.35$ | (0.5, 1.5) |

Table 5: Parameter settings of the experiment reported in Section 4.3 Figure 7. The five random seeds for each setting are {43, 37, 13, 51, 71}, and the means and standard deviations of the test accuracy among the five seeds are reported.

| Index | Dataset | Model | Method | HT-SR Metric | Initial learning rate $\eta_0$ | Test Acc | scaling ratio $(s_1, s_2)$ |
|---|---|---|---|---|---|---|---|
| 0 | | ResNet18 | TB | SpectralNorm | 0.05, 0.1, 0.15 | $77.83 \pm 0.21, 78.30 \pm 0.32, 78.27 \pm 0.25$ | (0.5, 1.5) |
| 1 | | ResNet18 | TB | AlphaWeighted | 0.05, 0.1, 0.15 | $78.18 \pm 0.27, 78.67 \pm 0.17, 78.48 \pm 0.24$ | (0.5, 1.5) |
| 1 | | ResNet18 | TB | PL_Alpha_Hill | 0.05, 0.1, 0.15 | $78.48 \pm 0.27, 78.97 \pm 0.29, 78.69 \pm 0.11$ | (0.5, 1.5) |
| 2 | | ResNet34 | TB | SpectralNorm | 0.05, 0.1, 0.15 | $78.25 \pm 0.16, 78.71 \pm 0.15, 78.92 \pm 0.28$ | (0.5, 1.5) |
| 3 | | ResNet34 | TB | AlphaWeighted | 0.05, 0.1, 0.15 | $78.36 \pm 0.39, 78.87 \pm 0.34, 78.83 \pm 0.23$ | (0.5, 1.5) |
| 3 | CIFAR100 | ResNet34 | TB | PL_Alpha_Hill | 0.05, 0.1, 0.15 | $79.36 \pm 0.18, 79.89 \pm 0.15, 79.09 \pm 0.64$ | (0.5, 1.5) |
| 4 | | VGG16 | TB | SpectralNorm | 0.025, 0.05, 0.1 | $73.58 \pm 0.19, 74.29 \pm 0.16, 74.17 \pm 0.28$ | (0.5, 1.5) |
| 5 | | VGG16 | TB | AlphaWeighted | 0.025, 0.05, 0.1 | $73.97 \pm 0.22, 74.19 \pm 0.11, 74.42 \pm 0.31$ | (0.5, 1.5) |
| 5 | | VGG16 | TB | PL_Alpha_Hill | 0.025, 0.05, 0.1 | $74.40 \pm 0.31, 74.96 \pm 0.15, 74.94 \pm 0.16$ | (0.5, 1.5) |
| 6 | | VGG19 | TB | SpectralNorm | 0.025, 0.05, 0.1 | $72.34 \pm 0.26, 72.91 \pm 0.35, 73.04 \pm 0.39$ | (0.5, 1.5) |
| 7 | | VGG19 | TB | AlphaWeighted | 0.025, 0.05, 0.1 | $72.85 \pm 0.16, 73.41 \pm 0.17, 73.33 \pm 0.21$ | (0.5, 1.5) |
| 7 | | VGG19 | TB | PL_Alpha_Hill | 0.025, 0.05, 0.1 | $73.47 \pm 0.16, 73.77 \pm 0.43, 73.40 \pm 0.38$ | (0.5, 1.5) |

Table 6: Parameter settings of the experiment reported in Section 4.3 Figure 8. The five random seeds for each setting are {43, 37, 13, 51, 71}, and the means and standard deviations of the test accuracy among the five seeds, the means and standard deviations of the computation time of using TB among the 10 times are reported.

| Index | Dataset | Model | Method | PL fitting method | Initial learning rate $\eta_0$ | Test Acc | Computation Time (sec) | scaling ratio $(s_1, s_2)$ |
|---|---|---|---|---|---|---|---|---|
| 0 | | ResNet18 | TB | Goodness-of-fit | 0.1 | $78.59 \pm 0.21$ | $8.20 \pm 0.53$ | (0.5, 1.5) |
| 1 | | ResNet18 | TB | Fix-finger | 0.1 | $79.06 \pm 0.22$ | $7.24 \pm 0.74$ | (0.5, 1.5) |
| 1 | | ResNet18 | TB | Median | 0.1 | $78.97 \pm 0.29$ | $1.14 \pm 0.04$ | (0.5, 1.5) |
| 2 | | ResNet34 | TB | Goodness-of-fit | 0.1 | $79.13 \pm 0.21$ | $16.45 \pm 0.48$ | (0.5, 1.5) |
| 3 | | ResNet34 | TB | Fix-finger | 0.1 | $79.64 \pm 0.22$ | $15.13 \pm 1.05$ | (0.5, 1.5) |
| 3 | CIFAR100 | ResNet34 | TB | Median | 0.1 | $79.89 \pm 0.15$ | $2.27 \pm 0.06$ | (0.5, 1.5) |
| 4 | | VGG16 | TB | Goodness-of-fit | 0.05 | $74.46 \pm 0.24$ | $8.54 \pm 0.10$ | (0.5, 1.5) |
| 5 | | VGG16 | TB | Fix-finger | 0.05 | $74.48 \pm 0.20$ | $8.45 \pm 0.59$ | (0.5, 1.5) |
| 5 | | VGG16 | TB | Median | 0.05 | $74.96 \pm 0.15$ | $1.37 \pm 0.05$ | (0.5, 1.5) |
| 6 | | VGG19 | TB | Goodness-of-fit | 0.05 | $73.36 \pm 0.16$ | $11.48 \pm 0.15$ | (0.5, 1.5) |
| 7 | | VGG19 | TB | Fix-finger | 0.05 | $73.52 \pm 0.16$ | $11.15 \pm 0.79$ | (0.5, 1.5) |
| 7 | | VGG19 | TB | Median | 0.05 | $73.77 \pm 0.43$ | $1.85 \pm 0.05$ | (0.5, 1.5) |

# E    Comparison with more baselines

In Figure 17, we provide additional results by comparing `TempBalance` with `LAMB` and `Adam`. We found that our method outperforms both baseline methods. Furthermore, we also found that the Adam-based methods do not provide better results than the `SGD` baseline with cosine annealing (`CAL`) in our experiment setting, which was mentioned in Section 4.2. For `Adam`, we searched the initial learning rate over $\{0.00005, 0.0001, 0.001, 0.01, 0.1\}$, and we used $\epsilon = 10^{-8}$. For `LAMB`, we searched the initial learning rate over $\{0.005, 0.01, 0.02\}$, and we used $\epsilon = 10^{-6}$. Both methods used weight decay $5.0 \times 10^{-4}$, $\beta_1 = 0.9$, $\beta_2 = 0.999$, learning rate decay with cosine annealing. Each experiment was conducted with five random seeds.

We also discuss the difference between `TempBalance` and these two types of learning rate scheduling.

- **Compared to layer-wise learning rate scheduling (e.g., `LARS`)**: `TempBalance` uses a more precise model quality metric, `PL_Alpha_Hill` from HT-SR Theory, to enhance the performance of deep models during training. This "shape-based" metric estimates the shape of the eigenspectrum of weight matrices. In contrast, `LARS` uses a "norm-based" metric, such as the layer-wise gradient norm. A recent study in HT-SR [22] has shown that the shape-based metrics surpasses norm-based ones in assessing model quality and performance. Figure 3 confirms that our method outperforms the layer-wise scheduler `LARS` in test accuracy.
- **Compared to parameter-wise learning rate scheduling (e.g., `Adam`)**: Similarly, our method employs the "shape-based" metric `PL_Alpha_Hill` to improve the generalization, an approach not incorporated in traditional parameter-wise methods.

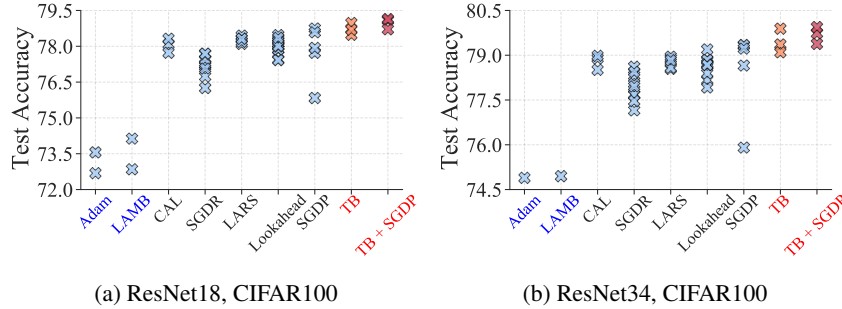

(a) ResNet18, CIFAR100          (b) ResNet34, CIFAR100

Figure 17: **(Comparison with additional baselines).** Comparing our method, `TempBalance` (TB), with other baselines such as parameter-wise learning rate schedulers `Adam` and `LAMB`, using ResNet18/34 trained on CIFAR100. Each cross represents the mean test accuracy of five random seeds.

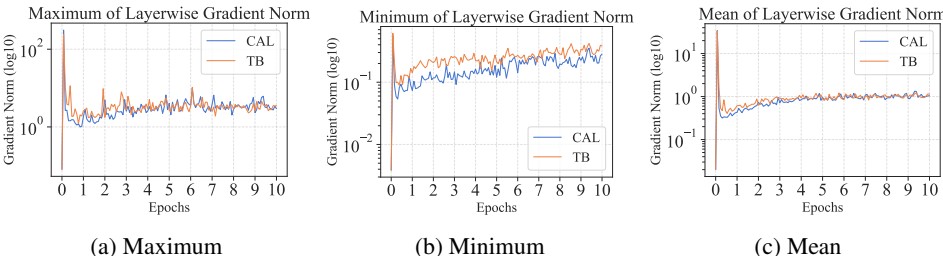

(a) Maximum          (b) Minimum          (c) Mean

Figure 18: **(Layerwise gradient norm during training).** From left to right: maximum, minimum, and mean of the layerwise gradient norm at every 30 iterations for the first 10 epochs. ResNet18 on CIFAR-100.

# F    Does addressing other training issues lead to `TempBalance`'s improvement?

We discuss whether the improvement from the proposed method, `TempBalance`, is due to indirectly addressing another fundamental training issue that could distort the ESD, specifically the gradient

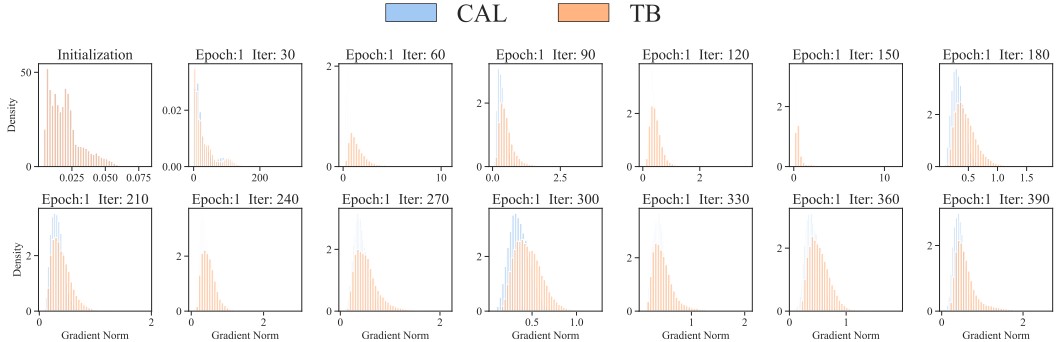

Figure 19: **(Histogram of gradient norm distribution during first epoch).** ResNet18 on CIFAR-100.

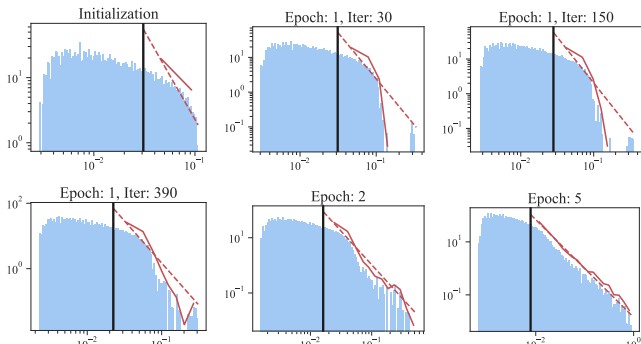

Figure 20: **(Impact of large rank-1 updates on ESD).** Large rank-1 updates result in the spikes of ESD, observed exclusively during the first epoch. From the second epoch onward, the ESD exhibits a heavy-tailed distribution. ResNet18 on CIFAR-100.

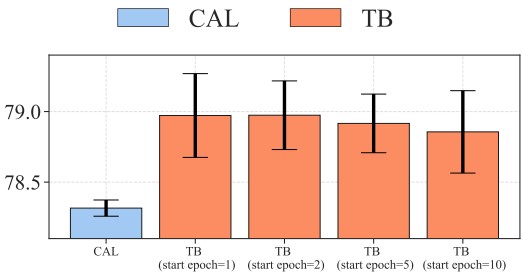

Figure 21: **(Varying the starting epoch of applying `TempBalance` (TB)).** Postponing the usage of `TempBalance` to Epochs 2, 5, and 10 doesn't affect the performance of `TempBalance` (originally starting from Epoch 1).

magnitude excursions [80] (explosion/vanishing). This discussion further strengthens the connection between our method and the HT structure, as discussed in the Sections 1, A, and B.

We first summarize the questions and the corresponding primary findings, with subsequent detailing of our experiment and supporting results.

- **Does gradient excursion exist?** We discovered that gradient explosion does exist, but it is confined to the first epoch out of a total of 200 training epochs, leading us to believe it does not significantly impact the test accuracy. We observed no gradient vanishing.
- **Does the observed gradient explosion impact the estimation of `PL_Alpha_Hill`?** We discovered that the large rank-1 updates resulting from the gradient explosion do indeed

affect the ESD as well as the `PL_Alpha_Hill` estimation. However, this effect is again restricted to the first epoch.
- **Does** `TempBalance` **boil down to addressing gradient explosion?** We found that postponing the use of `TempBalance` until the epoch when neither the gradient explosion nor the `PL_Alpha_Hill` estimation is affected does not compromise the test accuracy.

To support the above answers, we conducted three experiments. We discuss the setup of these experiments first and then analyze the results.

- (Figures 18, 19) We aim to detect gradient excursion by tracking the gradient norm across layers during training. We examine the model every 30 iterations over the first 10 epochs, calculating the $L_2$ norm of each gradient update across layers using the training batches of size 128. This produces an empirical gradient norm distribution with a total sample size of update numbers $\times$ layer numbers. Figure 18 presents the maximum/minimum/mean of the distribution, while Figure 19 visualizes these distributions for several iterations of Epoch 1.
- (Figure 20) We aim to assess the impact of gradient explosion on `PL_Alpha_Hill` estimation by monitoring the ESD. Figure 20 examines the change of the ESD of a single weight matrix over several iterations, tracked in the experiment depicted in Figures 18 and 19.
- (Figure 21) We aim to see if `TempBalance` enhances generalization by implicitly addressing gradient explosion. Since the gradient explosion and its effect on `PL_Alpha_Hill` estimation only transpire in the first epoch, we postpone the starting epoch of `TempBalance` to Epochs 2, 5, and 10 and see if it affects the test accuracy.

Our answers to the above questions are supported by the results obtained from the three experiments:

- **First question (Figure 18 and 19)**: We observed that the notable exploding gradients only occur in the initial 200 iterations of the first epoch. In Figure 18, we pinpoint a singular peak of maximum gradient norm within the first epoch. This aligns with the abnormal distribution with a large gradient norm in the subfigure of Figure 19 titled "Epoch 1, iteration 30."
- **Second question (Figure 20)**: Note that large rank-one updates have been studied in random matrix theory, which manifests as a "bulk+spike" pattern. This has been analyzed in, e.g., Theorem 2.13 of [94]. Figure 20 shows this "bulk+spike" pattern, but only in the first epoch. The ESD exhibits a heavy-tail distribution in subsequent epochs, suggesting the influence of rank-one updates is limited.
- **Third question (Figure 21)**: Delaying the application of `TempBalance` until after the first epoch does not adversely affect the test accuracy. Figure 21 illustrates that applying `TempBalance` from Epochs 2, 5, and 10 results in test performance comparable to when `TempBalance` is applied from Epoch 1. Since the gradient explosion only occurs in the first epoch and its effect on `PL_Alpha_Hill` estimation diminishes after this, the effectiveness of `TempBalance` does not rely on addressing gradient explosion or biased `PL_Alpha_Hill` estimation from large rank-one updates.
- **Third question**: We compare `TempBalance` with the baseline method `LARS`, which uses gradient norms to determine layer-wise learning rates in combating gradient vanishing/explosion issues. As illustrated in Figure 4, `TempBalance` outperforms `LARS` in terms of generalization performance.

# G   Corroborating results on other tasks

We provide corroborating results of applying `TempBalance` to two different tasks: object detection (OD) and language modeling (LM). In both tasks, `TempBalance` consistently improves generalization, outperforming the baseline scheduler cosine annealing (`CAL`) when both are combined with `Adam`/`AdamW` optimizers.

For OD, we studied the PASCAL VOC2007 [95] dataset with YOLO series [96] pre-trained model. We compared `TempBalance` with the baseline scheduler `CAL` with both applied to `Adam`/`AdamW` optimizer. For both scheduler methods, we trained for 200 epochs with batch size 64, and we set the same hyperparameter for the optimizers: $\beta_1 = 0.9$, $\beta_2 = 0.999$, $\epsilon = 10^{-8}$, weight decay = $5.0 \times 10^{-4}$. We searched the initial learning rate for all methods among $\{7.5 \times 10^{-6}, 1 \times 10^{-5}, 2.5 \times 10^{-5}\}$. For metrics we use the COCO [97] version mean Average Precision (mAP, higher is better), which is calculated for 10 IOUs varying in a range of 0.5 to 0.95 with steps of 0.05. We

report the mean of mAP over five random seeds on the test set. We set the scaling factors $(s_1, s_2)$ of `TempBalance` to be $(0.6, 1.4)$.

Here are the experimental settings for LM. We studied the Penn Treebank (PTB) dataset [98] using a three-layer "tensorized transformer core-1" [99]. We compared `TempBalance` with the baseline scheduler `CAL` with both applied to `Adam` optimizer. For both scheduler methods, we trained the models for 40K iterations with a batch size of 120, and a dropout rate of 0.3. We searched the initial learning rate for baseline methods among $\{1.25 \times 10^{-4}, 2.5 \times 10^{-4}, 5 \times 10^{-4}, 1 \times 10^{-3}, 1.25 \times 10^{-3}, 2.5 \times 10^{-3}, 5 \times 10^{-3}\}$ for baseline `CAL`. The hyperparameters for Adam are $\beta_1 = 0.9$, $\beta_2 = 0.999$, $\epsilon = 10^{-8}$. The mean of perplexity (PPL, lower is better) across five random seeds on the test set is reported. We observed improved performance of `TempBalance` in this task when extending our hyperparameter search to include the scaling factors $(s_1, s_2) \in \{(0.5, 1.5), (1.0, 2.0)\}$, the power-law fitting hyperparameter $\lambda_{\min}$ index $k \in \{\frac{n}{2}, \frac{n}{1.25}\}$, and the `TempBalance` update interval over $\{10, 25, 50\}$ iterations.

Table 7: (a) Object Detection (OD): mean Average Precision (mAP) on PASCAL VOC 2007 using model Yolov8n. (b) Language Modeling (LM): test perplexity (PPL) on Penn TreeBank (PTB) using the tensorized transformer. `TempBalance` (TB) consistently outperforms the `CAL` in different tasks.

| CAL + Adam | TB + Adam | CAL + AdamW | TB + AdamW | | CAL + Adam | TB + Adam |
|---|---|---|---|---|---|---|
| 59.59 | **60.03** (+0.44) | 59.68 | **59.96** (+0.28) | | 49.94 | **47.30** (-2.64) |

| (a) OD, VOC2007, mAP (↑) | (b) LM, PTB, PPL (↓) |
|---|---|

We present additional results in Figure 22, showing the application of our method `TempBalance` to ResNet 101 on CIFAR-100, and we compare it with the baseline (CAL). We searched the initial learning rate among $\{0.05, 0.1, 0.15\}$ for both the baseline and our method. The results report the mean and standard deviation across five seeds. We found that `TempBalance` offers improvements for the larger ResNet101 model comparable to those observed for ResNet18/34, demonstrating its potential for larger models.

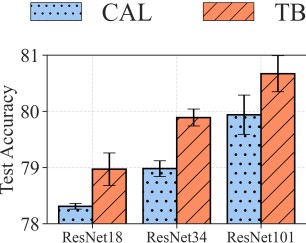

Figure 22: **(Applying the `TempBalance` (TB) to different sizes of ResNets)**. `TempBalance` consistently outperforms the baseline CAL method in the larger model ResNet101. The dataset is CIFAR100. Reporting mean/std over five random seeds.

## H   Analysis of computation overhead

We conducted a study on the computational overhead of `TempBalance`, demonstrating that our method is both applicable and scalable for large models. To do so, we conducted a scaling experiment to demonstrate that the computational cost remains low for different sizes of models. We recorded the duration of a single training epoch and the time taken to apply our method once. From this, we calculated the percentage increase in time when using `TempBalance` once per epoch, using this as an indicator of computational overhead. The experiment setup is based on ResNet-series on CIFAR100. We studied models of depth in $\{18, 34, 50, 101\}$ and ResNet18 models of width in $\{512, 768, 1024, 2048\}$. We report the mean and the standard deviation of the results over 10 runs. The test platform was one Quadro RTX 6000 GPU with Intel Xeon Gold 6248 CPU. The results are presented in Figure 23. Our findings reveal that the computational overhead remains low (less than 9%) even when applied to exceptionally wide or deep models (ResNet18 with width 2048 or ResNet101). The

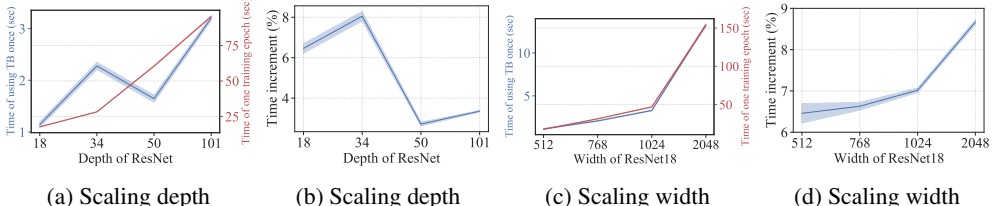

| (a) Scaling depth | (b) Scaling depth | (c) Scaling width | (d) Scaling width |

Figure 23: **(Computation overhead of** `TempBalance` **(TB) in scaling the model depth/width).**
(a)(c) Time duration (second) of one training epoch (blue) and using `TempBalance` once (red).
(b)(d) Time increment of using `TempBalance` once per epoch. The dataset is CIFAR100, reporting
mean/std over 10 epochs. The computational overhead of using `TempBalance` remains low (less
than 9%) even when applied to exceptionally wide or deep models.

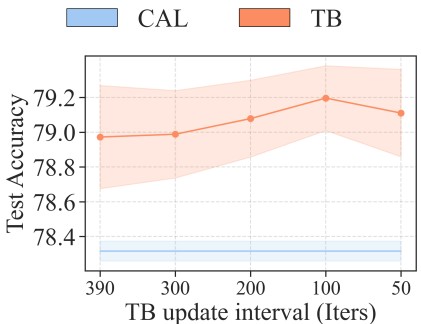

Figure 24: **(Varying the** `TempBalance` **(TB) update interval).** Reducing the update interval from
390 iters (used in the paper) brings mild improvement in test accuracy. Both use ResNet18 on
CIFAR100. Reporting mean/std over five random seeds.

computation overhead is not large because: 1) we select the efficient PL fitting method to obtain
`PL_Alpha_Hill`, which is demonstrated in Figure 8; and 2) the most computation-intensive part of
our method is SVD decomposition, which we have optimized using GPU implementation and batch
processing.

We conducted an experiment on reducing the update interval of the learning rate schedule to see
if it affects the test accuracy of `TempBalance`. Figure 24 shows the experiments conducted with
ResNet18 on CIFAR-100. We reduce the update interval from 390 iterations used in our paper
(equivalent to one epoch) to 300, 200, 100, and 50. We observed that there indeed exists a trade-off
between the computation time and test accuracy, but reducing the update interval only brings mild
improvement.

