# OpenReview forum: "Temperature Balancing, Layer-wise Weight Analysis, and Neural Network Training"
_NeurIPS.cc/2023/Conference — NeurIPS 2023 spotlight_

### Official Review · Reviewer_Gitq · 2023-06-30

**Soundness:** 2 fair
**Presentation:** 3 good
**Contribution:** 2 fair
**Rating:** 7
**Confidence:** 4

**Summary:**

This paper proposes TempBalance for temperature balancing based on the theory of heavy-tail self-regularization (HT-SR), which is a simple yet effective layer-wise policy applicable to general global temperature allocations in deep learning regularization. This paper proposes learning rate balancing across layers, which has received less attention compared to global or parameter-wise learning rate assignment. The HT-SR motivated capacity control metrics characterize the layers to achieve maximum temperature balance during model training, resulting in improved performance during testing. Extensive experiments show that TempBalance significantly outperforms ordinary SGD and carefully tuned spectral norm regularization, as well as a number of state-of-the-art optimizers and learning rate schedulers.

**Strengths:**

1. The article proposes a simple yet effective layer-wise learning rate schedule TempBalance based on HT-SR theory.
2. The article compares TempBalance with SGD and SNR on various training tasks, including different network architectures (such as ResNet, VGG, WideResNet), different datasets (such as CIFAR10, CIFAR100, SVHN, TinyImageNet), and extensive ablation studies (such as varying widths, depths, initial learning rates and HT-SR layer-wise metrics).
3. TempBalance outperforms a range of state-of-the-art optimizers and learning rate schedulers, and maintains stable performance over SGD baselines when the model size changes.
4. The designed algorithm is simple and easy to understand, and the paper is well-written.

**Weaknesses:**

1. The algorithm implementation is too simple and lacks theoretical innovation.
2. The motivation is not clear enough, and it does not explain why it is necessary to design layer-wise learning rate schedule strategy based on HT-SR theory.
3. The paper only conducts experiments on classification tasks and does not conduct experimental validation on downstream tasks or other areas.

**Questions:**

1. What are the advantages of layer-wise learning rate schedule strategy based on HT-SR theory over other layer-wise learning rate schedule strategies? What are the advantages over other global and parameter-wise learning rate schedule strategies?
2. Optimizers are closely related to the model architecture, e.g. Transformers typically use the second-order optimizers, while CNNs typically use the first-order optimizers. In this paper, we do not analyze and experimentally verify this common phenomenon, and only use CNNs in combination with first- and second-order optimizers for classification tasks, which may lead to "We do not find them to provide better results than the SGD baseline with cosine annealing". Can more extensive experimentation on this issue be done when conditions permit?

**Limitations:**

This article provides necessary discussions on the limitations and future directions that can be explored.

---

> ### Author Rebuttal · Authors · 2023-08-09
>
> ## Experiments on other areas
> In Table 6 of the rebuttal PDF, we provide experiments of applying our method TempBalance (TB) to two different tasks: object detection and language modeling. In both tasks, TB consistently improves generalization, outperforming the baseline scheduler cosine annealing (CAL) when both are combined with Adam/AdamW optimizers.
>
> Here are experimental settings for object detection. We utilized the PASCAL VOC2007 [1] dataset and the YOLO-v8n [2] model with the pre-trained weights from the official website. The baseline methods are CAL combined with Adam/AdamW, while our method is TB with Adam/AdamW. For both methods, we trained for 200 epochs with batch size 64 and set the same hyperparameter for the optimizers: weight decay = $5.0×10^{−4}$, $\beta_{1}$ = 0.9, $\beta_{2}$ = 0.999, $\epsilon$ = $10^{−8}$. We searched the initial learning rate for all methods among {$7.5×10^{-6}$, $1×10^{-5}$, $2.5×10^{-5}$}. We report the mean and standard deviation of mean Average Precision (mAP, higher is better) over five random seeds on the test set.
>
> Here are experimental settings for language modeling. We studied the Penn Treebank (PTB) dataset [3] using a three-layer "tensorized transformer core-1"[4]. We compared our method TB with the baseline scheduler CAL with both applied to the Adam optimizer. For both methods, we trained the models for 40000 iterations with a batch size of 120 and a dropout rate of 0.3. We searched the initial learning rate for all methods among {0.000125, 0.00025, 0.0005, 0.001, 0.00125, 0.0025, 0.005}. The hyperparameters for Adam are $\beta_{1}$ = 0.9, $\beta_{2}$ = 0.999, $\epsilon$ = $10^{−8}$. The mean and standard deviation of perplexity (PPL, lower is better) across five random seeds on the test set are reported.
>
> ## Experiments with Transformer and Adam
> We agree that optimizers are closely related to the architecture and using the Adam with Transformer instead of CNN is a more suitable experimental setup. In our rebuttal PDF Table 6 (b), we trained a tensorized transformer using Adam. Further details are in "Experiments on other areas." Our results show that TB outperforms the Adam baseline with the CAL scheduler, demonstrating its compatibility with Transformers and Adam.
>
> ## Advantages over different learning rate scheduling
> 1. **Compared to layer-wise learning rate scheduling (e.g., LARS):** Our method utilizes a more precise generalization metric, the alpha metric from heavy-tailed self regularization (HT-SR) theory, to enhance the generalization performance of deep models during training. This "shape-based" metric estimates the shape of the eigenspectrum of weight matrices. In contrast, LARS uses a "norm-based" metric, such as the layer-wise gradient norm. A recent study in HT-SR [5] has shown that the shape-based metric alpha surpasses norm-based ones in assessing model generalization performance. Figure 4 of the submitted paper confirms that our method outperforms the layer-wise scheduler LARS in test accuracy.
>
> 2. **Compared to parameter-wise learning rate scheduling (e.g., Adam):** Similarly, our method employs the "shape-based" metric alpha to improve the generalization, an approach not incorporated in traditional parameter-wise methods. Our updated results in Figure 18 of the rebuttal PDF confirm that our method outperforms the parameter-wise scheduler Adam and LAMB in test accuracy. Moreover, our updated experiments in Table 6 show that combining our method with Adam/AdamW further improves the generalization.
>
> Here are the experimental setups for Figure 18. For Adam, we searched the initial learning rate over {0.00005, 0.0001, 0.001, 0.01, 0.1}, we used $\epsilon$ = $10^{−8}$. For LAMB, we searched the initial learning rate over {0.005, 0.01, 0.02}, and used $\epsilon$ = $10^{−6}$. Both used a weight decay of $5.0×10^{−4}$, $\beta_{1}$ of 0.9, $\beta_{2}$ of 0.999, learning rate decay with cosine annealing. Each experiment was conducted with five random seeds. All other hyperparameters were consistent with those described in the paper. The experimental details of Table 6 can be found in our response titled "Experiments on other areas".
>
> ## Lack of theoretical innovation and simple algorithm design
> We wish to emphasize that this is the first study to design learning rate schedulers based on the heavy-tailed self regularization (HT-SR) theory. We draw the theoretical insights from HT-SR, noting that the weight matrices of layers in well-trained models typically exhibit a heavy-tailed eigenspectrum. We propose to use the Hill estimator to quantify the heavy-tailed pattern of each neural network layer, utilizing this value for more steady layer-wise learning rate scheduling. This theory-guided method is novel, efficient in improving test accuracy, and straightforward to implement, as affirmed by the reviewer in the "Strengths" section.
>
> ## Unclear motivation: why design layer-wise learning rate schedule strategy based on HT-SR theory
> The heavy-tailed self regularization (HT-SR) theory suggests that the empirical spectral densities of weight matrices in a well-trained neural network typically display a heavy-tailed pattern. This heavy-tailed characteristic can indicate the quality of each layer, signifying whether it is overtrained or undertrained. Such theoretical insights motivate us to balance the overtrained and undertrained levels of different layers by designing a layer-wise learning rate schedule based on the heavy-tailed pattern measurement. Specifically, we assign higher learning rates to undertrained layers (as indicated by less pronounced heavy-tailed patterns) and lower rates to overtrained layers (indicated by more pronounced heavy-tailed patterns). We dynamically monitor these measurements and adjust the scheduling throughout the training process.
>
> ## Reference
> [1] Everingham et al, 2010.
>
> [2] Redmon et al, 2016.
>
> [3] Mikolov et al, 2011.
>
> [4] Ma et al, 2019.
>
> [5] Martin and Michael, 2021.

---

### Official Review · Reviewer_QLkn · 2023-07-05

**Soundness:** 3 good
**Presentation:** 3 good
**Contribution:** 3 good
**Rating:** 7
**Confidence:** 2

**Summary:**

The paper proposes a layer-wise learning rate scheduler that adapts the learning rate to the eigenvalue distribution of the Weight-covariance matrix. As explained in the appendix, output and weight covariance matrices as well as the Hessian and Fischer-Information matrix are closely related, and they often show a power-law decay in their eigenvalues. The hypothesis is that layers with a smaller exponent in the power-law-decay are more "overtrained" than those with larger values, and thus should receive smaller updates. This behavior is implemented by interpolating learning rates linearly by (fitted) decay-exponents. Experiments show an improved generalization performance over base-line methods for a variety of standard computer-vision benchmarks and (CNN) architectures.

**Strengths:**

The paper follows an interesting and less explored approach to understanding the training dynamics of networks. The observation of an emergence of power laws as such is fascinating, and studying the connection to training dynamics adds an interesting and novel perspective. Further, the empirical results are encouraging: It seems that the proposed criterion for adaptive training does indeed improve generalization performance, which is an unexpected, non-trivial result that on its own raises some eyebrows (i.e., should be discussed and explored).

The paper is well-written and in particular the discussion of the background in the appendix provides an interesting and insightful read.

**Weaknesses:**

My main concern with the submission in its current form is that it is light on the analytical side. In short, the paper does not try to explain why there is a connection between training success and the proposed criterion/schedule and there is a risk of overlooking unmodelled or indirect effects: In my experience, the training process of multi-layer networks is affected by a mix of numerical and fundamental issues that are usually difficult to separate.

A very significant problem are for example gradient magnitude excursions (explosion/vanishing). For networks that contain batch normalization layers (or most variants thereof), the standard He initialization leads to exploding gradients at initialization. This causes early layers to "learn" at an exponentially larger rate than later layers, but the effect vanishes over time. It seems likely, that large rank-1 updates from initially exploding gradients distort the estimation of the decay coefficients ("alpha"), and likely in the "correct" direction of dampening exploding layers. Residual architectures still suffer from this effect within each stack in each residual block (but to a lesser degree). In contrast, non-normalized networks such as the traditional VGG suffer from vanishing gradients for other reasons. Then, there is the problem of concentration of the overall singular value spectrum for stacks of random matrices (higher-order powers of Wigner spectra), which again affects the effective rank of the updates.

In order to understand better if and how the proposed method could (or does) work, it would be useful to take such effects into account, for example by measuring gradient magnitudes over layers, or by a theoretical model. This could inform the reader better with respect to the  technical justification the proposed scheme and might help removing (or justifying) ad-hoc choices, such as linear interpolation of learning rates. It would also be helpful to carefully consider architectural aspects such as normalization and residual connections, and (at least) monitor the layer-wise statistics of gradient magnitudes and decay coefficients in order to understand better what is going on.

There are some smaller issues that could be improved, such as experimenting on a more diverse set of architectures and data sets and comparing against a larger set of base-line schedulers. In the former case, I would consider the current effort sufficient (as computational costs are skyrocketing easily, and the effect does seem quite pronounced already), in the latter case a more comprehensive study (maybe only for one or a few representative examples) could help in solidifying the result further.

Overall, I found this paper very interesting and the results are unexpectedly strong. The only reason for my slightly negative overall assessment is that (for a paper at NeurIPS) more effort could be put into understanding the results better (empirically and/or analytically).

**Questions:**

Are the new results state-of-the-art in terms of generalization performance (compared with other LR-schedulers)? That does not need to be the case to have an interesting paper, but it would be good to know which characteristics of other approaches reach similar goals.

Conceptually: Would it be possible that the whole effect observed boils down to implicitly addressing exploding or vanishing gradients?

EDIT: Post rebuttal, I have raised my score to accept (and soundness and contribution accordingly), as the response of the authors shows very clearly that the new approach has effects beyond gradient magnitude excursions, which was my main concern in terms of empirical evidence for the effect described. I would encourage the authors to clarify this in a revision of the paper.

**Limitations:**

My main concern in terms of limitations is the possibility that other primary effects case the spectral criterion to trigger as a secondary feature. This could be excluded by additional experiments (or even some theory). One could also discuss limitations of the empirical study a bit more in detail, but I would think that the limitations are obvious to an attentive reader, so there would be no serious issues in this regard (I would not understand this as a "new technique" paper but rather a paper that explores a novel methodological approach, not yet arriving at a deployable method).

---

> ### Author Rebuttal · Authors · 2023-08-09
>
> ## Does TempBalance (TB) boil down to addressing gradient excursions
> We first summarize the reviewer's questions and our primary responses, with subsequent detailing of our experiment and supporting results.
>
> 1. **Does gradient excursion exist?**
> We discovered that gradient explosion does exist, but it is confined to the first epoch out of a total of 200 training epochs, leading us to believe it does not significantly impact the test accuracy. We observed no gradient vanishing.
> 2. **Does the observed gradient explosion impact the estimation of alpha?**
>   We discovered that the large rank-1 updates resulting from the gradient explosion indeed affect the Empirical Spectral Density (ESD) as well as the alpha estimation. However, this effect is again restricted to the first epoch.
> 3. **Does TB boil down to addressing gradient explosion?**
>    We found that postponing the use of TB until the epoch when neither the gradient explosion nor the alpha estimation is affected does not compromise the test accuracy.
>
> To support the above answers, we conducted three experiments (see figures in the rebuttal material). We discuss the setup of these experiments first and then analyze the results.
>
> * (Figures 12,13) We aim to detect gradient excursion by tracking the gradient norm across layers during training. We examine the model every 30 iterations over the first 10 epochs, calculating the $L_2$ norm of each gradient update across layers using the training batches of size 128. This produces an empirical gradient norm distribution with a total sample size of update numbers $\times$ layer numbers. Figure 12 presents the maximum/minimum/mean of the distribution, while Figure 13 visualizes these distribution for several iterations of Epoch 1.
> * (Figure 14) We aim to assess the impact of gradient explosion on alpha estimation by monitoring the ESD. Figure 14 examines the change of the ESD of a single weight matrix over several iterations, tracked in the experiment depicted in Figures 12 and 13.
> * (Figure 15 (a)) We aim to see if TB enhances generalization by implicitly addressing gradient explosion. Since the gradient explosion and its effect on alpha estimation only transpire in the first epoch, we postpone the starting epoch of TB to Epochs 2, 5, and 10 and see if it affects the test accuracy.
>
> Our responses to the questions are supported by the results obtained from the three experiments:
> 1. **First question (Figure 12 and 13):** We observed that the notable exploding gradients only occur in the initial 200 iterations of the first epoch. In Figure 12, we pinpoint a singular peak of maximum gradient norm within the first epoch. This aligns with the abnormal distribution with a large gradient norm in the subfigure of Figure 13 titled "Epoch 1, iteration 30".
> 2. **Second question (Figure 14):** Note that large rank-one updates have been studied in random matrix theory, which manifests as a "bulk+spike" pattern. This has been analyzed in, e.g., Theorem 2.13 of [1]. Figure 14 shows this "bulk+spike" pattern, but only in the first epoch. The ESD exhibits a heavy-tail distribution in subsequent epochs, suggesting the influence of rank-one updates is limited.
> 3. **Third question (Figure 15 (a)):** Delaying the application of TB until after the first epoch does not adversely affect the test accuracy. Figure 15 (a) illustrates that applying TB from Epochs 2, 5, and 10 results in test performance comparable to when TB is applied from Epoch 1. Since the gradient explosion only occurs in the first epoch and its effect on alpha estimation diminishes after this, the effectiveness of TB does not rely on addressing gradient explosion or biased alpha estimation from large rank-one updates.
> 4. **Third question**: We compare TB with the baseline method LARS, which uses gradient norms to determine layer-wise learning rates in combating gradient vanishing/explosion issues. As illustrated in Figure 4 of the submitted paper, TB outperforms LARS in terms of generalization performance.
>
> ## Light on the analytical side
> 1. Our method is founded on the HT-SR theory, detailed in both the introduction and Appendix A. This explains: **(1) our use of alpha for better generalization:** The paper [2] showed that modern neural networks' Empirical Spectral Densities (ESDs) typically exhibit a heavy-tail distribution. Alpha, the decay coefficient of ESD, effectively gauges generalization. [3] provided a rigorous bound for this, and [4] pinpointed an optimal alpha value close to 2. **(2) our approach to learning rate schedule based on alpha:** Based on insights from [2] and [5], we set layer-wise learning rates according to alpha values, as a higher learning rate decreases alpha. Thus, layers with larger alpha get higher learning rates. Our linear interpolation design for learning rate assignment was based on its better performance over other designs (e.g., square root, log2, step). See Figure 16 in the rebuttal PDF.
>
> 2. The success of TB and its relationship with HT-SR are elucidated in Appendix B (Figures 10, 11). It reveals that TB effectively controls the shape of the ESDs. Compared to CAL, TB consistently attains a more concentrated distribution, with both the mean and median approaching the theoretically optimal PL_Alpha_Hill value of 2 [4]. This is consistently observed across various settings.
>
> ## Smaller issues: comparison to other schedulers
>
> Figure 4 in our paper demonstrates our method's advantage over multiple baselines. Our rebuttal's Figure 18 further confirms its consistent improvement over parameter-wise schedulers such as Adam and LAMB. Also, our rebuttal PDF Table 6 reveals our method's performance gains in object detection and language modeling.
>
> ## Reference
> [1] Couillet and Liao, 2022
>
> [2] Martin and Mahoney, 2021
>
> [3] Simsekli et al, 2020
>
> [4] Bartlett et al, 2020
>
> [5] Gurbuzbalaban et al, 2020

---

> > ### Comment · Reviewer_QLkn · 2023-08-15
> >
> > Dear Authors,
> >
> > thanks for the very detailed reply and extensive additional analysis. The additional findings and explanations indeed resolve the open problems I saw previously.
> >
> > The issue of gradient magnitude excursions is usually limited to the first few steps of optimization in networks with batch normalization, a the normalization layer will counteract weight excursions from large gradient updates and effectively freeze the affected layers. That your method still provides an improvement after the first epoch is a clear indication (in my perception) that it does more than "just" counteracting gradient magnitude differences. I also overlooked the LAMB results already in the paper (LAMBs per layer adjustment avoid such problems completely, but the new method is still better).
> >
> > I would correspondingly adjust my score and recommend mentioning the differentiation to "just" exploding gradients in the final version (maybe providing some of the new results in a suitable way). If possible, it would also be good to add a few more sentences on the HT-theory (maybe in the appendix) to help readers less familiar with the background.
> >
> > Thanks again for the detailed feedback!

---

> > > ### Author Response · Authors · 2023-08-15
> > > **Thank you for the response**
> > >
> > > Thank you for the constructive comments. We are glad that our rebuttal helps answer your questions and clarify that our method is more than mitigating the issue of gradient explosion/vanishing. We agree that addressing gradient magnitude excursions is vital for improving training. We will ensure that these discussions on gradient magnitude excursions, along with the details on HT-SR theory and the results on LAMB, are highlighted in the final version.

---

### Official Review · Reviewer_MZpk · 2023-07-09

**Soundness:** 3 good
**Presentation:** 2 fair
**Contribution:** 2 fair
**Rating:** 6
**Confidence:** 2

**Summary:**

This paper propose TempBalance, an adaptive lr schedule that assigns lr to each layer based on its heavy-tail characterization. The authors estimate PL_Alpha, the exponent of the power law distribution that fits the heavy tail part of the empirical spectral density, for the weight at each layer. They propose to assign a higher lr to the layer with larger PL_Alpha and lower lr to the layer with smaller PL_Alpha as a larger PL_Alpha often indicates a layer is under-trained while a smaller PL_Alpha often indicates a layer is over-trained.



**Strengths:**

1. The proposed TempBalance is novel in that it adjusts lr based on metrics from HT-SR theory, providing a new perspective from statistical mechanics of learning to neural networks optimization.
2. The experiments results demonstrate that TempBalance exhibits regularization effect and improves the generalization performance upon some existing lr schedulers, optimizers and spectral norm regularization methods.

**Weaknesses:**

1. Eq. 2: The authors design the relation between the lr and the value of PL_Alpha_Hill to be linear, which seems a bit arbitrary to me. Have the authors tried other designs?
2. The authors do not provide a convergence analysis.
3. Eq. 2: TempBalance requires the computation of eignevalues of the weight at each layer, inducing some computational overheads. Therefore, it can be difficult to scale TempBalance to models with larger widths and depths. The authors reduce the lr update frequency to alleviate the cost, but this might compromise the model performance.
4. Missing baseline: LAMB [1]. No hyper-parameter study on $s_1$ and $s_2$.
5. The authors only evaluate on small models. It would be good if the authors can provide an experiment on a ResNet-101 to demonstrate its potential for larger models.

[1] You, Yang, et al. "Large batch optimization for deep learning: Training bert in 76 minutes." arXiv preprint arXiv:1904.00962 (2019).



**Questions:**

1. Could authors provide a visualization of the layerwise lr with some insights on how lr varies across layers and how layerwise lr changes through training?
2. See weakness.

**Limitations:**

The authors have address the limitations.

---

> ### Author Rebuttal · Authors · 2023-08-09
>
> ## Other design of learning rate assignment?
> Our selection of the linear interpolation design for learning rate assignment in our TempBalance (TB) method was based on its superior performance in our ablation study, as provided in rebuttal PDF Figure 16.
>
> We evaluated three alternative learning rate assignment functions: Square root (Sqrt), Log2, and Step:
> * Sqrt : $f_t(i)=\eta_t\frac{\sqrt{\alpha_t^i}}{\frac{1}{L} \sum_{j=1}^{L}\sqrt{\alpha_t^j}}$,
> * Log2: $f_t(i)=\eta_t\frac{log(\alpha_t^i)}{\frac{1}{L} \sum_{j=1}^{L}log(\alpha_t^j)}$,
> * Step: For layer $i$ with $k$-th minimum alpha among all the layers,
>    $f_t(i)=  \eta_t (s_1 + (k-1)\frac{s_2 - s_1}{L-1}) $ .
>
> Here, $\eta_t$ denotes the base global learning rate at epoch $t$, $(s_1, s_2)$ represents the minimum and maximum learning rate scaling ratios relative to $\eta_t$, $\alpha_t^i$ is the PL_Alpha_Hill estimate of the layer $i$ at epoch $t$, and $L$ is the total number of model layers. All these notations are consistently used in the main paper.
>
> As depicted in Figure 16, our method, TB, surpasses the other designs when tested on VGG and ResNet architectures on CIFAR100. All hyperparameters are consistent with the main paper. Each experiment was conducted with five random seeds.
>
> ## Computation problem
> 1. **Is our method difficult to scale to large models?**
>     * In rebuttal PDF Figure 17, we conducted a scaling experiment to show that our method is applicable to large models. The experiment setup is based on ResNet-series on CIFAR100. We studied models of depth in {18, 34, 50, 101} and ResNet18 models of width in {512, 768, 1024, 2048}. For each model size, we recorded the duration of a single training epoch and the time taken to apply our method once. From this, we calculated the percentage increase in time when using TB once per epoch, using this as an indicator of computational overhead. Our findings reveal that the computational overhead remains low (less than 9%) even when applied to exceptionally wide or deep models (ResNet18 with width 2048 or ResNet101). We report the mean and the standard deviation of the results over 10 runs. The test platform was one Quadro RTX 6000 GPU with Intel Xeon Gold 6248 CPU.
>     * The computation overhead is not large because the most computation-intensive part of our method is SVD decomposition, which we have optimized using GPU implementation and batch processing.
>
> 2. **Does reducing the SVD computation compromise the test accuracy?**
> We conducted an experiment on reducing the update interval of learning rate schedule to see if it affects the test accuracy of TB. Figure 15 (b) shows the experiments conducted with ResNet18 on CIFAR-100. We reduce the update interval from 390 iters used in our paper (equivalent to one epoch) to 300, 200, 100, and 50. We observed that there indeed exists a trade-off between the computation time and test accuracy, but reducing the update interval only brings mild improvement.
>
> ## Missing baseline and hyperparameter study
> 1. **Missing baseline.** In rebuttal PDF Figure 18, we provide additional results by comparing our method with Adam and LAMB. We found that our method outperforms both baseline methods. Furthermore, we also found that the Adam-based methods do not provide better results than the SGD baseline with cosine annealing (CAL) in our experiment setting, which was mentioned in line $267-268$ in the paper. For Adam, we searched the initial learning rate over {0.00005, 0.0001, 0.001, 0.01, 0.1}, and we used $\epsilon$ = $10^{−8}$. For LAMB, we searched the initial learning rate over {0.005, 0.01, 0.02}, and we used $\epsilon$ = $10^{−6}$. Both methods used weight decay = $5.0×10^{−4}$, $\beta_{1}$ = 0.9, $\beta_{2}$ = 0.999, learning rate decay with cosine annealing. Each experiment was conducted with five random seeds.
>
> 2. **Missing hyperparameter study.** In Figure 19, we provide additional results of a hyperparameter study on $(s_1, s_2)$, in which we consider five different settings for $(s_1,s_2)$: $[(0.5,1.5), (0.6,1.4),(0.7,1.3), (0.8,1.2), (0.9,1.1)]$. We run tasks on CIFAR100 with four VGG and ResNet architectures, each with five random seeds. Our results show that a larger learning rate scaling range $(0.5,1.5)$ performs best. This hyperparameter setting is the default setting used in our paper. All hyperparameters are consistent with those described in the paper.
>
> ## Experiments on ResNet 101
> We present additional results in rebuttal PDF Figure 20, showing the application of our method (TB) to ResNet 101 on CIFAR-100, and we compare it with the baseline (CAL). We searched the initial learning rate among {0.05, 0.1, 0.15} for both the baseline and our method. The results report the mean and standard deviation across five seeds. We found that TB offers improvements for the larger ResNet101 model comparable to those observed for ResNet18/34, demonstrating its potential for larger models. We used the same hyperparameters as those for ResNet18/34 in Appendix C.
>
> ## Visualization of layer-wise learning rate
> In rebuttal PDF Figure 21, we visualize the layer-wise learning rates for ResNet 18/34 trained on CIFAR-100. We report the learning rate (or alpha) every epoch throughout the 200-epoch training duration.
> 1. **How does the learning rate vary across layers?** We observed a correlation between the layer-wise learning rate and the layer-wise alpha distribution: layers with larger alphas are allocated larger learning rates, whereas those with smaller alphas receive smaller learning rates.
> 2. **How does the layer-wise learning rate evolve during training?** The variations in layer-wise learning rates closely reflect shifts in the layer-wise alpha distribution. Initially, the alpha distributes uniformly across layers but eventually converge to a layer-wise pattern where earlier layers have smaller alphas and later layers have larger ones.

---

> > ### Comment · Reviewer_MZpk · 2023-08-16
> > **Thanks for your response**
> >
> > The new experiment results and analysis have addressed my concerns. My score has been updated.

---

> > > ### Author Response · Authors · 2023-08-16
> > >
> > > Thank you for the valuable feedback. We will ensure the new results and analysis are incorporated into the updated version.

---

### Official Review · Reviewer_zFJE · 2023-07-14

**Soundness:** 4 excellent
**Presentation:** 4 excellent
**Contribution:** 3 good
**Rating:** 9
**Confidence:** 3

**Summary:**

The paper proposes a way of modulating the learning rate, independently for each layer, when training deep networks via gradient descent.
This modulation keeps the average learning rate (over all layers) on a predefined path (e.g., cosine decay), but "balances" it according to the relative training stage of each layer: layers comparatively "overtrained" get a smaller scaling factor, layers comparatively "undertrained" get a larger one.
That comparison is done by leveraging the theory of "heavy-tail self-regularization" (HT-SR), estimating the heavy-tail characteristics of the empirical spectrum density (ESD), the spectrum of eigenvalues of weight correlation matrix, specifically the alpha coefficient of a power law fitting the heavy tail.

A wide range of experiments on small-scale image datasets, across architectures (ResNet, VGG) and variants (width and depth) show this method leads to better generalization error, compared to single-learning-rate optimizers, and can be further improved when combined with spectral norm regularization (SNR).

**Strengths:**

Originality
--------------
Layer-wise scaling factors for learning rates is an under-explored area. Exploring it by applying the results of theoretical models of deep network training is novel, and welcome.

Quality
----------
The paper presents extensive experiments supporting the proposed methods, and the papers conclusion. They feature a breadth of variants for the VGG and ResNet architectures, and hyper-parameters that have been carefully tuned.

Experiments have been replicated with 5 different seeds, and error bars reported. Experimental settings are clearly reported in the appendix. Overall really solid experimental validation.

Clarity
---------
The paper is clearly organized, and explained really clearly the theoretical bases it builds on, existing algorithms, as well as the proposed new method.

Significance
-----------------
A method that improves the generalization of existing architectures is extremely interesting, especially when the computation overhead is reasonable. This method could be quite impactful, either directly or through further improvements or further research in a similar direction.

**Weaknesses:**

No major weaknesses, just a few limitations.

1. It would have been interesting to see one series of experiments on larger-scale data (full ImageNet?), or maybe non-image data.
2. No explicit comparison with parameter-wise scaling schemes (e.g., a variation of Adam)
3. No mention of the algorithmic complexity, or overhead of computing these scales, before the penultimate paragraph.

**Questions:**

Clarification questions
1. l. 129, $\lambda_\mathrm{min}$ as the medium (median?) of the ESD. Is that what is shown on Figure 1? It was unclear to me how $\lambda_\mathrm{min}$ was selected based on the figure.
2. Should $(s_1, s_2)$ be mentioned in the Hyperparameters section (l.235)?

Further information / curiosity

3. In structured architectures like ResNet, is there a pattern of under / overtraining, either within each block, or between them? For instance, are layers within a block usually at the same "stage", or do you see correlations between the first layers of each stage? This might suggest ways in which to share a scaling factor within a block, for instance.
4. Is there a usual way in which the $\alpha_t^i$ usually evolve during regular SGD training, or a pattern? If there is, is it disrupted or modified when using `TempBalance`?
5. Similarly, how do the $f_t(i)$ evolve through time? For instance, smoothness may indicate that increasing the frequency of update may not be beneficial, but instabilities may suggest the opposite.
6. Is there evidence that convergence with `TempBalance` could be faster or slower than SGD? Could it compensate for the overhead of computing `PL_Alpha_Hill`, or worsen it?


*Update after rebuttal*

I believe all my questions were addressed. The additional experiments are thorough and lead me to increase my score.

**Limitations:**

Properly addressed within the body of the article.

---

> ### Author Rebuttal · Authors · 2023-08-09
>
> ## Experiments with non-image data
> In Table 6 (b) of our rebuttal PDF, we present a new experiment using a language dataset. Our TempBalance (TB) method performs better than the baseline cosine annealing learning rate schedule (CAL) when both used the Adam optimizer for language modeling.
>
> Here are experimental settings for language modeling. We studied the Penn Treebank (PTB) dataset [1] using a three-layer "tensorized transformer core-1" [2]. We compared our method TB with the baseline scheduler CAL with both applied to the Adam optimizer. For both methods, we trained the models for 40000 iterations with a batch size of 120, and a dropout rate of 0.3. We searched the initial learning rate for all methods among {0.000125, 0.00025, 0.0005, 0.001, 0.00125, 0.0025, 0.005}. The hyperparameters for Adam are $\beta_{1}$ = 0.9, $\beta_{2}$ = 0.999, $\epsilon$ = $10^{−8}$. The mean and standard deviation of perplexity (PPL, lower is better) across five random seeds on the test set are reported.
>
> ## Explicit comparison with parameter-wise scaling schemes
> In Figure 18 of the rebuttal PDF, we compared our method with parameter-wise learning rate schedulers including Adam and LAMB. We show that our method outperforms them with ResNet18/34 on CIFAR100.
>
> Here is the experimental setup. For Adam, we searched the initial learning rate over {0.00005, 0.0001, 0.001, 0.01, 0.1}, and used $\epsilon$ = $10^{−8}$. For LAMB, we searched the initial learning rate over {0.005, 0.01, 0.02}, and used $\epsilon$ = $10^{−6}$. Both methods used weight decay = $5.0×10^{−4}$, $\beta_{1}$ = 0.9, $\beta_{2}$ = 0.999, learning rate decay with cosine annealing. All results were obtained by running five random seeds.
>
> ## $\lambda_{min}$ as the median of the ESD (l.129)?
> In line 129, we state that the $\lambda_{min}$ is fixed as the median of all eigenvalues in the Empirical Spectral Density (ESD), represented by the black vertical line in Figure 1. The histogram plot's log scales on both axes might make this less intuitive. We thank the reviewer for pointing out the typo, and we will fix and clarify it in the revised version of the paper.
>
> ## Should $(s_1,s_2)$ be mentioned in the hyperparameters section (l.235)?
> We have listed the settings for $(s_1,s_2)$ for each experiment of the main paper in Appendix C's Table 1-5 (Hyperparameter settings) under the last column. We will mention it in line 235 in the revised draft.
>
> ## No mention of computation overhead
> We provide an additional study on the computation overhead of our TB method, with results presented in rebuttal PDF Figure 17. We conducted a scaling experiment to demonstrate that the computational cost remains low for different sizes of ResNet models. Our findings reveal that the computational overhead remains low (less than 9%) even when applied to exceptionally wide or deep models (ResNet18 with width 2048 or ResNet101).
>
> The experiment setup is based on ResNet-series on CIFAR100. We studied models of depth in {18, 34, 50, 101} and ResNet18 model of width in {512, 768, 1024, 2048}. For each model size, we recorded the duration of a single training epoch and the time taken to apply our method once per epoch. From this data, we calculated the percentage increase in time when using TB once per epoch, using this as an indicator of computational overhead. We report the mean and standard deviation of the results over 10 runs. The test platform used was a Quadro RTX 6000 GPU with an Intel Xeon Gold 6248 CPU. We will be sure to include these discussions in the updated draft of the paper.
>
> ## Patterns of under/overtraining between ResNet blocks
> In the rebuttal PDF, Figure 21 (b, d) illustrate the layer-wise alpha of ResNet 18 and 34 during training. Note that the ResNet 18/34 architecture is organized into four stages, with each stage comprising multiple residual blocks. Our primary focus is on the blue curves, which represent the alpha value at the end of training. One can recognize the patterns between stages: earlier layers (closer to the input) tend to have a greater number of layers with lower alpha values (indicative of overtraining) compared to later layers. Within one stage of the network, the second convolutional layer typically exhibits a lower alpha compared to the first convolutional layer. We will be sure to include more discussions about over/undertraining patterns in network architectures, as well as the future work on saving the computations by implementing a shared scaling factor for certain layers in the updated draft of the paper.
>
>
> ## How $\alpha_{t}^{i}$ and $f_{t}(i)$ evolve through training
> In Figure 22 of the rebuttal PDF, we present visualizations of the $\alpha_t^i$ (alpha) and $f_t(i)$ (learning rate) of two layers within the same ResNet18 during the training process.
>
> From Figure 22 (b, d), we can see that with the baseline CAL scheduler (blue curves), the earlier layer (index=1) achieves a smaller alpha value compared to the larger alpha value of the later layer (index=15). In contrast, our TB method (orange curves) narrows this gap, indicating our approach balances the undertraining/overtraining levels (as signified by alpha) of different layers. This is further corroborated by Figures 10 and 11 in the submitted paper, where our method consistently refines the layer-wise alpha distribution.
>
> Regarding the learning rate plots in Figure 22 (a, c), our TB method allocates a lower learning rate for earlier layers and a higher one for later layers than the baseline does. This leads to a more balanced alpha distribution between layers as mentioned above. Additionally, we noted instability in the learning rate curves during early training phases, while smoother transitions emerge in later phases.
>
> ## Comparison of convergence rate
> We observed that our method converges at the same rate as the SGD baseline, but it achieves higher test accuracy upon convergence.
>
> ## Reference
> [1] Mikolov et al, 2011.
>
> [2] Ma et al, 2019.

---

> > ### Comment · Reviewer_zFJE · 2023-08-16
> > **Thank you!**
> >
> > Thanks for the update and additional thorough experiments. I believe all my questions are addressed.

---

> > > ### Author Response · Authors · 2023-08-16
> > > **Thank you for the response**
> > >
> > > We appreciate your valuable feedback and will make sure to include the discussions and new experiments in the revised paper.

---

### Author Rebuttal · Authors · 2023-08-09

We want to thank all the reviewers for the constructive feedback, which helps us improve our paper. Please refer to the attached PDF for our new experiments and see below for our responses to each comment.

---

### Author Response · Authors · 2023-08-14
**A gentle reminder about the discussion phase**

Dear reviewers, thanks for your constructive feedback. Please tell us if we should include anything further in the revised draft. We are more than happy to clarify if anything is unclear.

---

### Decision · Program_Chairs · 2023-09-21

**Decision:**

Accept (spotlight)

**Comment:**

The paper proposes a theoretically-motivated procedure for automatically balancing learning rates across different layers of a neural network.  Experimental results demonstrate the empirical effectiveness of the layer-wise learning rates derived from the method.  After the rebuttal and discussion phase, all reviewers recommend accept, with some notable enthusiasm over the results achieved by the proposed optimizer.  The AC agrees with the reviewer consensus.